# HOTTEL ZONE PHYSICS-CONSTRAINED NETWORKS FOR FURNACES

## ABSTRACT

This paper investigates a novel approach to improve the temperature profile pre-
diction of furnaces in foundation industries, crucial for sustainable manufacturing.
While existing methods like the Hottel Zone model are accurate, they lack real-time
inference capabilities. Deep learning methods excel in speed and prediction but
require careful generalization for real-world applications. We propose a regulariza-
tion technique that leverages the Hottel Zone method to make deep neural networks
physics-aware, improving prediction accuracy for furnace temperature profiles.
Our approach demonstrates effectiveness on various neural network architectures,
including Multi-Layer Perceptrons (MLP), Long Short-Term Memory (LSTM),
Extended LSTM (xLSTM) and Kolmogorov-Arnold Networks (KANs). We also
discussion the data generation involved.

## 1 INTRODUCTION

Majority of economically relevant industries (automobiles, machinery, construction, household
appliances, chemicals, etc) are dependent on the Foundation Industries (FIs) that provide crucial and
foundational materials like glass, metals, cement, ceramics, bulk chemicals, paper, steel, etc. FIs are
heavy revenue and employment drivers, for instance, FIs in the United Kingdom (UK) economy are
worth £52B (EPSRC report), employ 0.25 million people, and comprise over 7000 businesses (IOM3
report). However, despite their economic significance, the FIs leverage energy-intensive methods
within their furnaces. This makes FIs major industrial polluters and the largest consumers of natural
resources across the globe. For example, in the UK, they produce 28 million tonnes of materials per
year, and generate 10% of the entire UK's $CO_2$ emissions (EPSRC report; IOM3 report). Similarly,
in China, the steel industry accounted for 15% of the total energy consumption, and 15.4% of the
total $CO_2$ emissions (Zhang et al., 2018; Liang et al., 2020). These numbers put a challenge for the
FIs in meeting our commitment to reduce net Green-House Gas (GHG) emissions, globally.

With a closer look at any process industry (e.g., steel industry), one can observe that at the core, lies
the process of conversion of materials (e.g., iron) into final products. This is done using a series of
unit processes (Yu et al., 2007) involving steps such as dressing, sintering, smelting, casting, rolling,
etc (see Qin et al. (2022) for an illustration). The equipment in such process industries operates
in high-intensity environments (e.g., high temperature), and has bottleneck components such as
reheating furnaces, which require complex restart processes post-failure. This causes additional labor
costs and energy consumption. Thus, for sustainable manufacturing, it is important to monitor the
temperature profile, and thus, the operating status of the furnaces. (Hu et al., 2019) have shown
promise in achieving notable fuel consumption reduction by reducing the overall heating time.

Yuen & Takara (1997) in their study, have proved the elegance and superiority of the Hottel Zone
method over counterparts to model the physical phenomenon of Radiative Heat Transfer (RHT) in
high-temperature processes. Hu et al. (2016) proposed a computational model workflow based on the
Hottel Zone method, and showed superiority over surrogate computational alternatives in terms of
predictive performance. However, none of these approaches are suitable for real-time inference in
modeling a furnace temperature profile. Deep Learning (DL) based neural network methods excel in
achieving superior predictive performance and speed. Nonetheless, their generalization capabilities
require special attention, particularly in critical real-world applications.

In our work, we propose to revisit the Hottel Zone method and devise a novel regularization technique
that could be used as a plug-and-play module to make a neural network physics-constrained (or

physics-aware) with regard to the underlying phenomena of high-temperature processes in furnaces. We show that for a time-step in a furnace, given a certain set of input entities, we could predict the desired output temperature entities more accurately (in terms of regression metrics) using our regularization technique, as opposed to using a vanilla neural network. We demonstrate the prowess of our proposal on different types of neural network architectures: Multi-Layer Perceptron (MLP) or feed-forward networks, sequential models such as Long Short-Term Memory (LSTM) based Recurrent Neural Networks (RNNs), as well as recently proposed Kolmogorov-Arnold Networks (KANs) and Extended LSTM (xLSTM).

This work makes two **key contributions**: **Tensor-based Reformulation and Physics-Aware Neural Networks**: We reformulate the Hottel Zone Method's Directed Flux Areas (DFAs) and Energy Balance (EB) equations in tensor format, enabling neural network training. We further introduce a novel regularization technique that imbues the network with physics-awareness. **Extensive Experimental Validation**: We comprehensively validate the proposed approach using various neural network architectures. To this end, we suggest a dataset and benchmarking protocol (details provided in Section A.8). A github repository is maintained at `https://github.com/` to facilitate real-time updates to the same as and when made.

Numerous real-world applications, including chemical reactors (Feng & Han, 2012), solar energy (Muhich et al., 2016; Marti et al., 2015), and 3D printing (Tran & Lo, 2018; Zhou et al., 2009), involve high-temperature processes exceeding $700°C$. These processes rely heavily on Radiative Heat Transfer (RHT) as a dominant mechanism alongside conduction and convection. Notably, RHT remains crucial for thermal transport even in vacuum conditions encountered in astronomical applications. We envision that our learnings could perhaps be extended to those applications with bespoke approaches.

Due to space constraints, we have limited the length of the introduction section. Please refer to Section A.1 for a more detailed discussion, particularly regarding the motivation behind our research.

## 2 RELATED WORK

In Section A.2, we provide a detailed discussion of related works. Due to space limitations, we will focus here on how our approach significantly differs from existing methods.

1. **View factor methods**: Existing methods Ebrahimi et al. (2013); Melot et al. (2011); Hu et al. (2018); Li (2005) simplify the modeling area and are geometry-specific. We propose a generic, geometry-agnostic model encompassing all exchange areas (radiation transfer interfaces).
2. **Neural network methods**: Existing methods Yuen (2009); Tausendschön & Radl (2021); García-Esteban et al. (2021); Zhai & Zhou (2020); Zhai et al. (2023); Halme Ståhlberg (2021); de Souza Lima et al. (2023); Liao et al. (2009); Hwang et al. (2019); Chen et al. (2022); Bao et al. (2023) often use simple MLPs, which lack generalization due to limited physics understanding. We introduce a Physics-constrained Neural Network (**PCNN**) framework that outperforms MLP and can be applied to other architectures like LSTM, KAN, xLSTM.
3. **Furnace temperature profiling**: Existing methods Kim & Huh (2000); Kim (2007); Jang et al. (2010); Tang et al. (2017); Nguyen et al. (2014); Hu et al. (2017); Ban et al. (2023); Li et al. (2023); Zanoli et al. (2023); Yu et al. (2022) focus on specific regions, while our method targets complete furnace temperature profiling, including gas zones, furnace walls, and slab surfaces. Our utilized data is more holistic. Existing neural methods in this category also lack physics awareness.
4. **PINNs**: Compared to the existing body of Physics-Informed Neural Network (PINN) literature Raissi et al. (2019); Karniadakis et al. (2021); Drgoňa et al. (2021); Shen et al. (2023); Cai et al. (2021); Kim et al. (2022); Zhao et al. (2020); He et al. (2021); Boca de Giuli (2023); Han et al. (2023); Bünning et al. (2022); Park (2022); Wang et al. (2023); Lahariya et al. (2022); Jing et al. (2023), we propose a novel variant specifically designed for zone method based modeling in reheating furnaces. Our approach is the first to utilize physics-constrained regularizers based on the zone method for temperature prediction. It requires minimal data (input-output pairs) and makes no geometry assumptions. Our data creation method is holistic and unique, encompassing all exchange areas. Our method, as we

will see later, is based on a set of simultaneous equations to incorporate physics-awareness, and directly does not involve a differential equation. Thus, we call it a physics-constrained method, though PINN could be also used philosophically.

## 3 PROPOSED METHOD

### 3.1 BACKGROUND

The Hottel Zone method subdivides a furnace into zones (volumes and surfaces) to predict Radiative Heat Transfer (RHT). Volume and Gas (G) zone is used interchangeably. Surface (S) zones are of two types, SF: furnace and SO: obstacle (e.g., slabs that are heated). Each zone has a uniform temperature. Sets of Energy-Balance (EB) equations govern radiation exchange between zones, considering incoming and outgoing radiation fluxes. These equations are iteratively updated to obtain the entire furnace's temperature profile. Following are the **key concepts**:

1. **Total Exchange Areas (TEAs)**: Pre-computed values representing the total area for radiation exchange between zone pairs (SS: surface-surface, SG/GS: surface-gas, GG: gas-gas).
2. **Directed Flux Areas (DFAs)**: Derived from TEAs and used to calculate radiant exchange between zone pairs at each step of the zone method.
3. **Weighted Sum of Grey Gases (WSGG) model**: Handles non-grey gases by representing them as a mixture of grey gases and a clear gas.

### 3.2 EXCHANGE AREA CALCULATION

The first step in the Zone method involves computation of Exchange Factors (Yuen & Takara, 1997). The exchange factor among a pair of volume zones $V_i$ and $V_j$ is expressed as:

$$g_i g_j = \int_{V_i} \int_{V_j} \frac{k_i k_j e^{-\tau} dV_i dV_j}{\pi r^2} \tag{1}$$

Physically, it represents the energy radiated from $V_i$ and absorbed/ scattered by $V_j$. Here, $k$ denotes the respective extinction coefficient, $\tau$ is the optical thickness among differential volume elements $dV_i$ and $dV_j$, and $r = \sqrt{(x_i - x_j)^2 + (y_i - y_j)^2 + (z_i - z_j)^2}$. Now, let $\boldsymbol{n_i}$ and $\boldsymbol{n_j}$ respectively be unit normal vectors of $dA_i$ and $dA_j$ (corresponding to two surface zones $A_i$ and $A_j$). Then, the exchange factors $g_i s_j$ (between volume zone $V_i$ and surface zone $A_j$) and $s_i s_j$ (between surface zone $A_i$ and surface zone $A_j$), can be expressed as:

$$g_i s_j = \int_{V_i} \int_{A_j} \frac{k_i |\boldsymbol{n_j}.r| e^{-\tau} dV_i dA_j}{\pi r^3}; s_i s_j = \int_{A_i} \int_{A_j} \frac{|\boldsymbol{n_i}.r||\boldsymbol{n_j}.r| e^{-\tau} dA_i dA_j}{\pi r^4} \tag{2}$$

Numerical evaluation of the above equations being complex, has led to analytical approximations, by considering an enclosure as a cube-square system, i.e, by representing a volume as a cube, and a surface as a square. This facilitates the tabulation of a "generic" set of exchange factors, which are applicable for most practical industrial geometries, using an updated Monte-Carlo based Ray-Tracing (MCRT) algorithm (Matthew et al., 2014). To this end, such pre-computed generic values are refered to as Total Exchange Areas (TEA), and we denote them by: $\overline{G_i S_j}$, $\overline{S_i S_j}$, $\overline{G_i G_j}$ and $\overline{S_i G_j}$. Here, $\overline{S_i G_j} = \overline{G_i S_j}$. Note that throughout the text, G(or g) and S(or s) shall indicate terms corresponding to Gas/Volume, and Surface respectively.

### 3.3 INTRODUCING TENSOR NOTATIONS FOR HOTTEL ZONE METHOD BASED NEURAL NETWORK

To account for our formulation of a neural network based approach, we first introduce the following four tensors to collectively represent the above TEAs: $\boldsymbol{GS} \in \mathbb{R}^{|G| \times |S| \times |N_g|}$, $\boldsymbol{SS} \in \mathbb{R}^{|S| \times |S| \times |N_g|}$, $\boldsymbol{GG} \in \mathbb{R}^{|G| \times |G| \times |N_g|}$, $\boldsymbol{SG} \in \mathbb{R}^{|S| \times |G| \times |N_g|}$. Here, $|G|$, $|S|$ respectively denote the number of gas/ volume zones, and number of surface zones. In practice, $|N_g|$ gases representing real gas medium are used, and hence, a third dimension has also been used in the above tensors. As discussed above, TEAs are pre-computed constants, used as inputs to our model. Slightly abusing notations, we can refer to a TEA by considering only the first two dimensions (for a pair of zones).

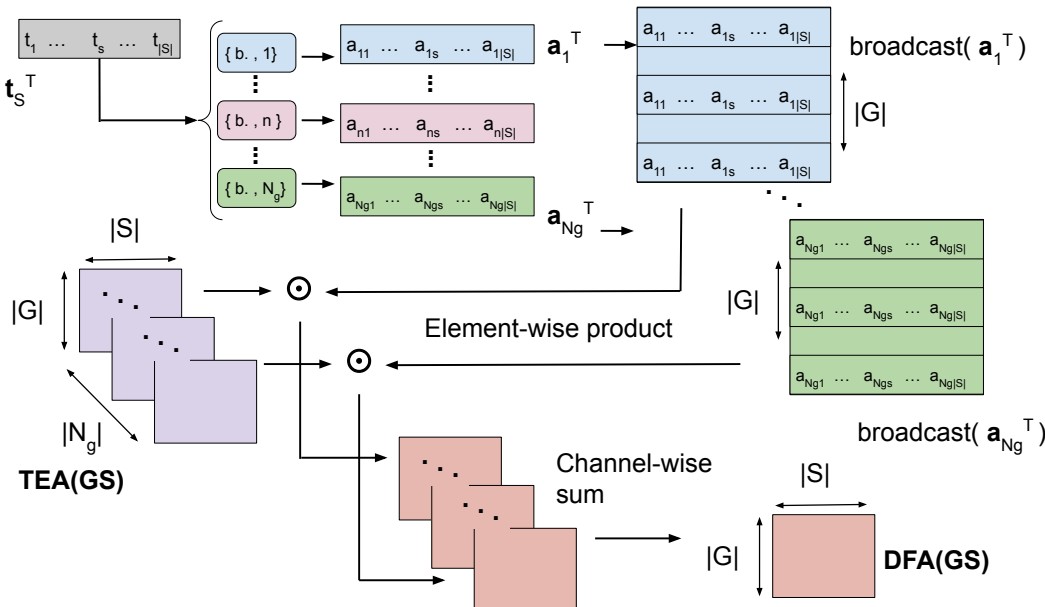

Figure 1: Derivation of matrix forms of the DFA terms (using GS as reference).

The next step is to compute the Radiation Exchange factors, or the Directed Flux Areas (DFA), considering radiating gas medium through a Weighted Sum of the mixed Grey Gases (WSGG) model (Hu et al., 2016):

$$\overset{\leftarrow}{G_iG_j} = \sum_{n=1}^{N_g} a_{g,n}(T_{g,j})(\overline{G_iG_j})_{k=k_n}; \overset{\leftarrow}{S_iS_j} = \sum_{n=1}^{N_g} a_{s,n}(T_{s,j})(\overline{S_iS_j})_{k=k_n} \tag{3}$$

$$\overset{\leftarrow}{G_iS_j} = \sum_{n=1}^{N_g} a_{s,n}(T_{s,j})(\overline{G_iS_j})_{k=k_n}; \overset{\leftarrow}{S_iG_j} = \sum_{n=1}^{N_g} a_{g,n}(T_{g,j})(\overline{S_iG_j})_{k=k_n} \tag{4}$$

Here, $\leftarrow$ indicates the direction of flow. $T_{g,j}$ and $T_{s,j}$ denote the temperatures for the $j^{th}$ volume and surface zones respectively, and are the values we want our model to predict (at each time step). Note that the collective representation of the DFAs can be expressed as: $\overset{\leftarrow}{GS} \in \mathbb{R}^{|G|\times|S|}$, $\overset{\leftarrow}{SS} \in \mathbb{R}^{|S|\times|S|}$, $\overset{\leftarrow}{GG} \in \mathbb{R}^{|G|\times|G|}$, $\overset{\leftarrow}{SG} \in \mathbb{R}^{|S|\times|G|}$. In Eq (3)-(4), the TEA terms correspond to a particular grey gas being used, for example, $(\overline{G_iG_j})_{k=k_n}$ represents the TEA $\overline{G_iG_j}$ with the $n^{th}$ gas.

WSGG is a method used to represent the absorptivity/ emissivity of real combustion products with a mixture of a couple of grey gases plus a clear gas, i.e, the number of grey gases is equal to $N_g - 1$.

For each gas indexed by $n$, we have a set of pre-computed correlation coefficients $\{b_{i+1,n}\}_{i=0}^{N_g}$ for both gas and surface related coefficients, and an absorption coefficient $k_{g,n}$. Then, the weighting coefficient $a_{g,n}(T_{g,j})$ (for gas-zone temperatures) and the weighting coefficient $a_{s,n}(T_{s,j})$ (for surface-zone temperatures) can be expressed as a $N_g^{th}$ order polynomial in $T_{g,j}$ (or $T_{s,j}$):

$$a_{g,n}(T_{g,j}) = \sum_{i=0}^{N_g} b_{i+1,n}T_{g,j}^i; a_{s,n}(T_{s,j}) = \sum_{i=0}^{N_g} b_{i+1,n}T_{s,j}^i \tag{5}$$

Using (3), (4, (5), and with GS as a reference, we make use of Figure 1 to illustrate the derivation of a compact matrix form for computing a DFA term efficiently for getting training samples of a neural network. Let, $(\overline{GS})_n$ be the $n^{th}$ slice of $GS$ along the third dimension, and $\boldsymbol{a}_n = \tilde{b}_n(\boldsymbol{t}_S)$. broadcast($\boldsymbol{a}_n^\top$) reshapes $\boldsymbol{a}_n^\top$ to the same dimension as $(\overline{GS})_n$, i.e., $\mathbb{R}^{|G|\times|S|}$. $\boldsymbol{t}_S \in \mathbb{R}^{|S|}$ is a vector containing all the surface zone temperatures (in a time step), such that its $j^{th}$ entry $\boldsymbol{t}_S(j) = T_{s,j}$. The $j^{th}$ entry $\boldsymbol{a}_n(j)$ of $\boldsymbol{a}_n \in \mathbb{R}^{|S|}$ is computed using the function $\tilde{b}_n$ with the correlation coefficients $\{b_{i+1,n}\}_{i=0}^{N_g}$ as the parameters, and by following eq (5). We can also assume similar vector containing all gas zone temperatures (in a time step) $\boldsymbol{t}_G \in \mathbb{R}^{|G|}$, with $j^{th}$ entry $\boldsymbol{t}_G(j) = T_{g,j}$.

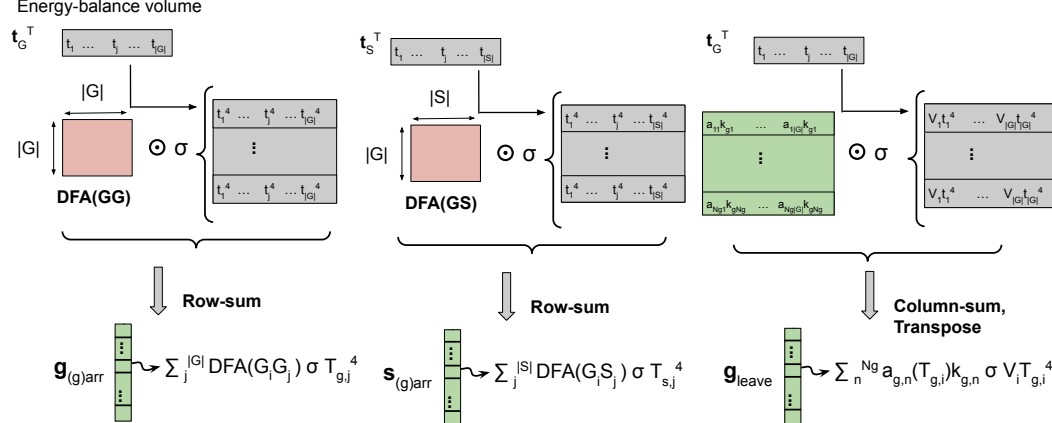

Figure 2: Derviation of the matrix forms of the EBV equations for physics based regularizers.

Then, the **DFA terms related to gas-zone temperatures** can be expressed as:

$$\overleftarrow{GS} = \sum_{n=1}^{N_g} (\overline{GS})_n \odot \text{broadcast}(\boldsymbol{a}_n^\top); \overleftarrow{GG} = \sum_{n=1}^{N_g} (\overline{GG})_n \odot \text{broadcast}(\tilde{b}_n(\boldsymbol{t}_G)^\top). \tag{6}$$

and, the **DFA terms related to surface-zone temperatures** can be expressed as:

$$\overleftarrow{SS} = \sum_{n=1}^{N_g} (\overline{SS})_n \odot \text{broadcast}(\tilde{b}_n(\boldsymbol{t}_S)^\top); \overleftarrow{SG} = \sum_{n=1}^{N_g} (\overline{SG})_n \odot \text{broadcast}(\tilde{b}_n(\boldsymbol{t}_G)^\top). \tag{7}$$

### 3.4 ENERGY-BALANCE BASED PHYSICS-REGULARIZATION

With the above DFA terms at our disposal, we can compute the gas/volume and surface zone temperatures at each time step of furnace operation by respectively using Energy-Balance Volume (EBV) and Energy-Balance Surface (EBS) equations. EBV and EBS are a set of simultaneous equations to capture the governing physics of RHT Hu et al. (2016). Figure 2 visually illustrates computation of the terms $\boldsymbol{g}_{(g)arr}$, $\boldsymbol{s}_{(g)arr}$ and $\boldsymbol{g}_{leave}$ involved in the EBV equation to compute the gas zone temperatures of a time step.

Let, $\boldsymbol{g}_{(g)arr} \in \mathbb{R}^{|G|}$ be a vector whose $i^{th}$ entry represents the amount of radiation arriving at the $i^{th}$ gas zone from all the other gas zones, $\boldsymbol{s}_{(g)arr} \in \mathbb{R}^{|G|}$, a vector whose $i^{th}$ entry represents the amount of radiation arriving at the $i^{th}$ gas zone from all the other surface zones, $\boldsymbol{g}_{leave} \in \mathbb{R}^{|G|}$, a vector whose $i^{th}$ entry represents the amount of radiation leaving the $i^{th}$ gas zone, and $\boldsymbol{h}_g \in \mathbb{R}^{|G|}$ a heat term. Also, let $T_{g,j}$ (or $T_g$) and $T_{s,j}$ (or $T_s$) denote the $j^{th}$ gas and surface zone temperatures respectively. Then, following EBV equations, the $i^{th}$ entries of $\boldsymbol{g}_{(g)arr}$, $\boldsymbol{s}_{(g)arr}$, $\boldsymbol{g}_{leave}$ and $\boldsymbol{h}_g$ can be computed as:

$$\boldsymbol{g}_{(g)arr}(i) = \sum_{j}^{|G|} \overleftarrow{G_i G_j} \sigma T_{g,j}^4; \qquad \boldsymbol{s}_{(g)arr}(i) = \sum_{j}^{|S|} \overleftarrow{G_i S_j} \sigma T_{s,j}^4$$

$$\boldsymbol{g}_{leave}(i) = \sum_{n}^{|N_g|} a_{g,n}(T_{g,i}) k_{g,n} \sigma V_i T_{g,i}^4 \quad \boldsymbol{h}_g(i) = -(\dot{Q}_{conv})_i + (\dot{Q}_{fuel,net})_i + (\dot{Q}_a)_i + \boldsymbol{q}_i \tag{8}$$

Here, the constants (known apriori) $(\dot{Q}_{conv})_i$, $(\dot{Q}_{fuel,net})_i$, and $(\dot{Q}_a)_i$ respectively denote the convection heat transfer, heat release due to input fuel, and thermal input from air/ oxygen. An enthalpy vector $\boldsymbol{q} \in \mathbb{R}^{|G|}$ is computed using the flow-pattern obtained via polynomial curve fitting during simulation. $\sigma$ is the Stefan-Boltzmann constant, $V_i$ is volume of $i^{th}$ gas zone.

Let, $\boldsymbol{s}_{(s)arr} \in \mathbb{R}^{|S|}$, be a vector whose $i^{th}$ entry represents the amount of radiation arriving at the $i^{th}$ surface zone from all the other surface zones, $\boldsymbol{g}_{(s)arr} \in \mathbb{R}^{|S|}$, a vector whose $i^{th}$ entry represents

the amount of radiation arriving at the $i^{th}$ surface zone from all the other gas zones, $\boldsymbol{s}_{leave} \in \mathbb{R}^{|S|}$, a vector whose $i^{th}$ entry represents the amount of radiation leaving the $i^{th}$ surface zone, and $\boldsymbol{h}_s \in \mathbb{R}^{|S|}$ a heat term. Then, following EBS equations, the $i^{th}$ entries of $\boldsymbol{s}_{(s)arr}$, $\boldsymbol{g}_{(s)arr}$, $\boldsymbol{s}_{leave}$ and $\boldsymbol{h}_s$ can be computed as:

$$\boldsymbol{s}_{(s)arr}(i) = \sum_j^{|S|} \overleftarrow{\boldsymbol{S_i S_j}} \sigma T_{s,j}^4; \quad \boldsymbol{g}_{(s)arr}(i) = \sum_j^{|G|} \overleftarrow{\boldsymbol{S_i G_j}} \sigma T_{g,j}^4 \tag{9}$$

$$\boldsymbol{s}_{leave}(i) = A_i \epsilon_i \sigma T_{s,i}^4; \qquad \boldsymbol{h}_s(i) = A_i (\dot{q}_{conv})_i - \dot{Q}_{s,i}$$

For a surface zone $i$, the constants (known apriori) $A_i(\dot{q}_{conv})_i$ and $\dot{Q}_{s,i}$ respectively denote the heat flux to the surface by convection and heat transfer from it to the other surfaces. Here, $A_i$ is the area, and $\epsilon_i$ is the emissivity of the $i^{th}$ surface zone.

The calculated terms in the Energy-Balance (EB) equations represent the heat entering and leaving each zone. In simpler terms, these equations ensure an energy balance by placing all incoming heat terms on the left-hand side (LHS) and outgoing terms on the right-hand side (RHS). Leveraging these terms in an optimization framework allows us to minimize the difference between LHS and RHS. To achieve this, we introduce the following terms:

$$\boldsymbol{v}_g = (\boldsymbol{g}_{(g)arr} + \boldsymbol{s}_{(g)arr} - 4\boldsymbol{g}_{leave} + \boldsymbol{h}_g) \in \mathbb{R}^{|G|}$$
$$\boldsymbol{v}_s = (\boldsymbol{s}_{(s)arr} + \boldsymbol{g}_{(s)arr} - \boldsymbol{s}_{leave} + \boldsymbol{h}_s) \in \mathbb{R}^{|S|} \tag{10}$$

Here, $|G|/|S|$ denotes the number of Gas/ Surface zones. Intuitively, $\boldsymbol{v}_g$ and $\boldsymbol{v}_s$ are vector representatives corresponding to EBV and EBS. Let, $\lambda_{ebv}, \lambda_{ebs} > 0$ are hyper-parameters corresponding to $\mathcal{L}_{ebv}$ and $\mathcal{L}_{ebs}$, such that $\mathcal{L}_{ebv} = ||\text{normalize}(\boldsymbol{v}_g)||_2^2$ is our proposed regularizer term corresponding to the **EBV**. Similarly, $\mathcal{L}_{ebs} = ||\text{normalize}(\boldsymbol{v}_s)||_2^2$ is our proposed regularizer term corresponding to the **EBS**. We use: $\text{normalize}(\boldsymbol{v}) = \boldsymbol{v}/\max(\boldsymbol{v})$, where $\max(\boldsymbol{v})$ is the maximum value from among all components in $\boldsymbol{v}$.

The core idea is to leverage the Energy Balance (EB) equations, which represent well-established physical laws governing heat transfer in the furnace. These equations enforce a balance between incoming and outgoing heat for each zone. The vectors $\boldsymbol{v}_g$ and $\boldsymbol{v}_s$ capture the residuals between the incoming and outgoing heat terms in the EB equations for gas (g) and surface (s) zones, respectively. By minimizing the L2 norm of these residuals (after normalization), we are essentially penalizing the network for deviating significantly from the physical constraints imposed by the EB equations. This encourages the network to learn temperature profiles that adhere to these well-defined energy balances.

Minimizing the L2 norm encourages the network to drive all components of the residual vectors towards zero. The normalization step ensures all zones contribute equally to the penalty, regardless of their absolute temperature values. This prevents zones with naturally higher temperatures from dominating the regularization term.

### 3.5 Putting together the neural network objective

We now discuss the design of our final neural network. We formulate the objective in such a way that we can plug the above proposed regularizers in a standalone neural network architecture trained to regress output temperatures given a set of easily available input entities at each time step of a furnace operation. While starting the furnace operation, ambient temperatures are readily available (depicting the *initial state of the furnace*), along with walk interval, desired target set point temperatures. Then, based on the firing rates chosen for the burners of the furnace, there would be a resulting flow pattern in the furnace. This is a result of heat flow, and mass flow within the furnace (mass flow happens because of the slab movements, which need to be heated). This flow pattern would cause a change in the overall enthalpy, leading to a new temperature profile (*new state*) of the furnace, which can be measured by the resulting new gas and surface zone temperatures. These temperatures in turn could serve as input temperatures for the next step's prediction. For a more intuitive understanding of furnace operation, please refer Section A.8.

In a practical setup, a neural network deployed could expect to consume the previous step temperatures, firing rates, walk interval, and set point temperatures as inputs. The output could then be the new

temperatures, and the next firing rates as well. With input-output data $\mathcal{X}=\{(\boldsymbol{x}^{(i)}, \boldsymbol{y}^{(i)})\}_{i=1}^N$ acquired in this manner, we can estimate parameters $\theta$ of a neural network $f_\theta(.)$ by training it to predict $\boldsymbol{y}^{(i)}$ given $\boldsymbol{x}^{(i)}$, for all time step $i$, as:

$$\theta^* \leftarrow \arg\min_\theta \mathcal{L}_{sup} \tag{11}$$

Here, $\mathcal{L}_{sup} = \mathbb{E}_{(\boldsymbol{x}^{(i)}, \boldsymbol{y}^{(i)}) \in \mathcal{X}}[||\boldsymbol{y}^{(i)} - f_\theta(\boldsymbol{x}^{(i)})||_2^2]$ is a standard *supervised term for regression*. To make such a network physics-aware, all we need to do is include the above proposed terms $\mathcal{L}_{ebv}$ and $\mathcal{L}_{ebs}$ into the final objective. It should be noted that, in doing so, we do not need to make any architectural changes to the network in terms of inputs and outputs. Also, all auxiliary variables used in computation of (8) and (9) are only used during training of a physics-aware network, and are not required in the inference.

The regularization terms are computed using additional vectors as described earlier, influence the learning because they have the temperature terms in them. For example, in (10), $\boldsymbol{v}_g$ depends on gas zone temperatures $T_{g,j}$ via $\boldsymbol{g}_{(g)arr}, \boldsymbol{g}_{leave}$ in (8). While computing $\mathcal{L}_{ebv}$ we obtain the $T_{g,j}$ terms using the network output, which are associated with the computational graph and thus help the updates during back-propagation. On the other hand, $\boldsymbol{s}_{(g)arr}$ is associated with $T_{s,j}$ which are detached for back-propagation while updating gas zone temperatures.

Similarly, in (10), $\boldsymbol{v}_s$ depends on surface zone temperatures $T_{s,j}$ via $\boldsymbol{s}_{(s)arr}, \boldsymbol{s}_{leave}$ in (9). While computing $\mathcal{L}_{ebs}$ we obtain the $T_{s,j}$ terms using the network output, which are associated with the computational graph and thus help the updates during back-propagation. On the other hand, $\boldsymbol{g}_{(s)arr}$ is associated with $T_{g,j}$ which are detached for back-propagation while updating surface zone temperatures.

The overall physics-aware loss is formulated as:

$$\mathcal{L}_{total} = \mathcal{L}_{sup} + \lambda_{ebv}\mathcal{L}_{ebv} + \lambda_{ebs}\mathcal{L}_{ebs} \tag{12}$$

When calculating the physics-aware loss terms we detach certain temperature terms associated with one zone type (e.g., surface zone temperatures) during updates of the other zone type (e.g., gas zone temperatures). This prevents the network from altering these relationships unnaturally during backpropagation. As analogy, we can refer to a Teacher-Student Learning setup: Imagine the network learning from a teacher (the EB equations) that provides the correct temperature relationships. Detaching specific terms allows the network to focus on learning the mapping between furnace inputs and its own predicted zone temperatures, while still adhering to the guidance provided by the teacher (the EB equations) through the physics-aware loss terms. Algorithm 1 provides detailed steps of our proposed approach.

---

**Algorithm 1** Algorithm of the proposed method

---

1: **Input:** $\mathcal{X}=\{(\boldsymbol{x}^{(i)}, \boldsymbol{y}^{(i)})\}_{i=1}^N$, furnace configuration (set points and walk interval). $maxeps > 0$.
2: Initialize $\theta$, TEAs, $\lambda_{ebv}, \lambda_{ebs} > 0$.
3: Initialize $\boldsymbol{t}_G \in \mathbb{R}^{|G|}, \boldsymbol{t}_S \in \mathbb{R}^{|S|}$ with ambient temperatures, and firing rates.
4: **for** EN=1 **to** $maxeps$ **do**                                    ▷ EN: Epoch No.
5:     **for** i=1 **to** $N$ **do**                                        ▷ i: time step
6:         Compute DFAs $\overleftarrow{\boldsymbol{GG}}^{(t)}, \overleftarrow{\boldsymbol{GS}}^{(t)}, \overleftarrow{\boldsymbol{SG}}^{(t)}, \overleftarrow{\boldsymbol{SS}}^{(t)}$ using (6) and (7).
7:         Compute $\mathcal{L}_{ebv}$ using (8) and (10).
8:         Compute $\mathcal{L}_{ebs}$ using (9) and (10).
9:         Compute $L_{sup}$ using $\mathcal{X}$.
10:         $\theta^{(i)} \leftarrow \theta^{(i-1)} - \eta\nabla_\theta\mathcal{L}_{total}$                       ▷ Using (12)
11:     **end for**
12: **end for**
13: $\theta^* \leftarrow \theta^{N.maxeps}$
14: **return** $\theta^*$

---

## 4    EXPERIMENTS

In this section we report results on 11 datasets obtained using different configurations of a real-world furnace based on Hu et al. (2019) (details in Section A.8.3). Major objective of the experiments is

Table 1: Comparison of proposed methods on the N1-2 Dataset

| Dataset | N1-2 | | | | 965_1220_1250_750 | | | | | |
|---|---|---|---|---|---|---|---|---|---|---|
| Metric/ Method | MLP | PBMLP | LSTM | PBLSTM | DLSTM | PBDLSTM | KAN | PBKAN | xLSTM | PBxLSTM |
| RMSE tG (↓) | 113.4 | 35.6 | 33.0 | 26.7 | 117.1 | 32.4 | 24.3 | **22.6** | 130.6 | 29.3 |
| RMSE tS fur (↓) | 116.4 | 22.4 | 25.6 | **11.7** | 114.4 | 15.2 | 14.6 | 13.8 | 119.1 | 20.4 |
| RMSE tS obs (↓) | 106.9 | 43.4 | 61.1 | 66.5 | 109.3 | 67.4 | 35.1 | **33.6** | 139.8 | 45.4 |
| MAE tG (↓) | 89.5 | 28.2 | 27.4 | **16.9** | 100.9 | 27.2 | 21.4 | 19.9 | 129.1 | 26.8 |
| MAE tS fur (↓) | 96.2 | 17.8 | 21.5 | **9.9** | 101.1 | 20.1 | 14.3 | 13.8 | 118.6 | 19.5 |
| MAE tS obs (↓) | 79.9 | 29.6 | 39.4 | 31.4 | 86.9 | 44.4 | 29.8 | **29.3** | 136.3 | 39.8 |
| mMAPE fr (↓) | 176.6 | 58.5 | 29.5 | **23.5** | 201.0 | 26.2 | 44.2 | 32.6 | 200.8 | 27.8 |

Table 2: Comparison of proposed methods on the N2-1 Dataset

| Dataset | N2-1 | | | | 955_1190_1250_750 | | | | | |
|---|---|---|---|---|---|---|---|---|---|---|
| Metric/ Method | MLP | PBMLP | LSTM | PBLSTM | DLSTM | PBDLSTM | KAN | PBKAN | xLSTM | PBxLSTM |
| RMSE tG (↓) | 121.1 | 45.4 | 36.8 | 37.0 | 123.4 | 28.3 | 29.5 | **18.0** | 95.5 | 33.0 |
| RMSE tS fur (↓) | 123.8 | 27.6 | 29.5 | 28.9 | 120.5 | 18.7 | 20.7 | **8.8** | 80.6 | 24.9 |
| RMSE tS obs (↓) | 113.1 | 52.4 | 65.6 | 63.3 | 114.5 | 51.9 | 41.0 | **27.2** | 90.7 | 51.7 |
| MAE tG (↓) | 96.9 | 38.8 | 31.3 | 31.4 | 106.9 | 19.7 | 26.2 | **15.4** | 93.5 | 30.3 |
| MAE tS fur (↓) | 103.6 | 24.8 | 26.7 | 25.5 | 106.4 | 16.5 | 19.8 | **7.7** | 80.1 | 24.1 |
| MAE tS obs (↓) | 87.4 | 39.9 | 46.2 | 44.2 | 92.2 | **21.9** | 35.9 | 22.9 | 86.6 | 46.5 |
| mMAPE fr (↓) | 187.6 | 67.8 | 28.4 | 29.8 | 210.6 | **24.9** | 43.7 | 34.2 | 212.3 | 26.2 |

to consider different neural network architectures with and without our proposed regularizers (and keeping everything else constant). Any gains reported could be attributed to our proposed regularizers that seek to enhance the physics-awareness of a network. Results across all the 11 datasets are reported in Tables 6, 7, 8, 9.

For neural network architectures, we study following variants: MLP, LSTM, a stacked/deep LSTM (DLSTM) and recently proposed KAN and xLSTM. We use commonly used regression performance metrics such as RMSE and MAE for the temperature prediction. We also report MAPE additionally for predicting the next firing rates (MAPE is more suitable due to the range of values that firing rates take). A metric against each of the different entities has been reported. For example, RMSE tS fur denotes the average RMSE for all the furnace surface zone predictions, RMSE tS obs denotes the average RMSE for all the obstacle surface zone predictions, RMSE tG denotes the average RMSE for all the gas zone predictions. mMAPE fr indicates the performance on the firing rate predictions. For all metrics, a lower value indicates a better performance. All metrics are reported along the rows of a table, and the columns represent the different methods. For each row, the best performing metric corresponding to a method is shown in bold.

In Table 1 we report the performance of the architectures MLP, LSTM, DLSTM, KAN and xLSTM on the N1-2 dataset. We also report performances of PBMLP, PBLSTM, PBDLSTM, PBKAN and PBxLSTM, which are the Physics-Based (PB) variants of MLP, LSTM, DLSTM, KAN and PBxLSTM respectively. The green colored cells indicate that a PB variant has obtained a better performance than a vanilla variant without our proposed regularizers. Compared to the simpler MLP, we could see massive gains by the PBMLP.

The DLSTM (and xLSTM) variant possibly tends to overfit due to stacking of more LSTM layers, and performs worse compared to a vanilla LSTM model. Stacking LSTMs offered no advantage likely due to the data's inherent structure. Unlike language tasks that benefit from complex LSTM modeling with longer windows/time steps, zone-based method only requires capturing the relationship between the current state (s(i)) and the next (s(i+1)). Our data generation (details in Appendix) captures the relationship between current state (s(i)) and next state (s(i+1)), making complex LSTM architectures unnecessary. Initial experiments confirmed this, showing no significant improvement with longer windows compared to the simpler s(i), s(i+1) pairs. This aligns with Occam's razor - favoring simpler models with comparable performance.

However, when equipped with our regularizers, the PBDLSTM (and PBxLSTM) method obtains much better performance than the DLSTM (and xLSTM). The vanilla LSTM which performs better than the MLP and DLSTM, also obtains improvements after using the physics based regularizers, as indicated by the performance of PBLSTM. We also notice KAN to perform better than the base MLP (as observed in recent literature). In fact, the PBKAN variant performs the best among all methods at times.

In Table 2 we report performances of the same approaches on the N2-1 dataset. We observed similar conclusions: the PB variants were outperforming their vanilla variants (as shown by green), thus depicting the benefit of the proposed regularizers. In this case, we observed that the PBKAN method obtains the best performance among all.

Table 3: Comparison of proposed methods on average across the datasets.

| Dataset | Average | | | | | | | | | |
|---|---|---|---|---|---|---|---|---|---|---|
| Metric/ Method | MLP | PBMLP | LSTM | PBLSTM | DLSTM | PBDLSTM | KAN | PBKAN | xLSTM | PBxLSTM |
| RMSE tG ($\downarrow$) | 79.2 | 35.1 | 37.2 | 30.4 | 83.5 | 27.9 | 26.1 | **19.3** | 85.2 | 31.7 |
| RMSE tS fur ($\downarrow$) | 75.6 | 23.1 | 27.1 | 20.2 | 78.1 | 20.5 | 18.5 | **12.4** | 75.5 | 24.2 |
| RMSE tS obs ($\downarrow$) | 86.8 | 49.5 | 64.9 | 64.1 | 89.9 | 61.7 | 37.0 | **29.8** | 95.3 | 45.8 |
| MAE tG ($\downarrow$) | 62.2 | 29.1 | 29.7 | 23.8 | 70.9 | 22.4 | 23.8 | **16.8** | 83.4 | 29.5 |
| MAE tS fur ($\downarrow$) | 62.6 | 20.3 | 23.1 | 18.1 | 68.9 | 17.3 | 18.0 | **11.6** | 74.9 | 23.5 |
| MAE tS obs ($\downarrow$) | 62.5 | 33.9 | 40.7 | 38.6 | 65.3 | 36.0 | 33.4 | **25.7** | 90.9 | 40.5 |
| mMAPE fr ($\downarrow$) | 119.3 | 53.6 | 39.2 | 26.7 | 141.8 | **25.9** | 46.4 | 39.3 | 131.4 | 37.5 |

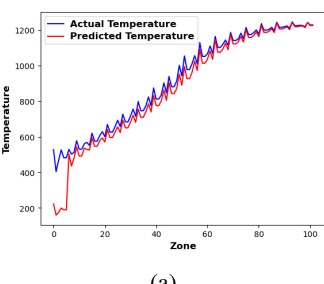 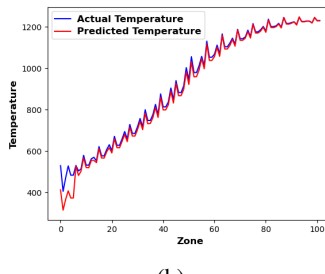

| (a) | (b) |

Figure 3: Plot of actual (blue) and predicted (red) temperatures (in $^\circ C$) across all obstacle surface zones using PBMLP. In (a) we omit previous furnace temperatures from the neural network input to show that performance degrades.

Difference in the datasets N1-2 and N2-1 comes by varying setpoint temperatures of the first and second control zones of the furnace. This shows that depending on the furnace configuration of the same geometry, the performance of a deep learning model may vary as the data distribution changes due to the difference in underlying physical entities. However, if equipped with physics based regularizers, we could make the network adhere to the governing laws, and get a reasonable predictive performance.

We further report on how the different methods perform across varying configurations or datasets on average, in Table 3. We observed similar performances, where the PB variants led to better performance. In Tables 6, 7, 8, 9 we report the performances of the compared approaches across all the 11 datasets. We noticed that not only the PB variants obtain a better performance throughout, they are also more stable across different datasets as indicated by their standard deviations.

In Figure 4 we plot the convergence of our PBMLP method. Losses with respect to all the individual terms converge well. In Figure 3 we report visual plots of actual and predicted temperatures for PBMLP. We also show that omitting previous temperatures from the neural network inputs leads to an worse performance, thus, highlighting the impact of a furnace state on the model performance. We conducted a sensitivity analysis of $\lambda_{ebv}$ and $\lambda_{ebs}$ in Figure 5, observing stable performance across values.

### 4.1 FINAL NOTE ON IMPACT OF ENERGY-BALANCE REGULARIZATION

Throughout the text, for all baseline methods in a column, the counterpart with the PB- prefix (eg, PBMLP, PBLSTM, PBDLSTM, PBKAN, PBxLSTM) indicates the usage of energy-balance regularization terms, and the green colored metrics all denote the consistent performance boost, as compared to the vanilla variants (eg, MLP, LSTM, DLSTM, KAN, xLSTM).

### 4.2 COMPARISON AGAINST RECENT STATE-OF-THE-ART (SOTA)

While we acknowledge the importance of contextualizing our work, we recognize that making direct comparisons is challenging due to the unique characteristics of our framework. Most existing methods in the literature focus on limited exchange areas in furnace temperature modeling. In contrast, our robust data generation framework encompasses the entire set of exchange areas, which is essential for accurate temperature profiling.

To facilitate meaningful comparisons, we relate our results to established baselines recognized as State-Of-The-Art (SOTA) techniques in settings similar to ours. Specifically, we evaluate the impact of our research by comparing our proposed Physics-Based (PB) variants against the following methods: i) MLRVPST (Bao et al. (2023)) and ii) PTDL-LSTM (de Souza Lima et al. (2023)), the

Table 4: Comparison of proposed methods on average across the datasets against recent SOTA.

| Dataset | Average | | | | | |
|---|---|---|---|---|---|---|
| Metric/ Method | MLRVPST (Bao et al. (2023)) | PTDL-LSTM (de Souza Lima et al. (2023)) | PBLSTM | PBDLSTM | PBKAN | PBxLSTM |
| RMSE tG ($\downarrow$) | 31.2 | 37.2 | 30.4 | 27.9 | **19.3** | 31.7 |
| RMSE tS fur ($\downarrow$) | 24.5 | 27.1 | 20.2 | 20.5 | **12.4** | 24.2 |
| RMSE tS obs ($\downarrow$) | 51.1 | 64.9 | 64.1 | 61.7 | **29.8** | 45.8 |
| MAE tG ($\downarrow$) | 28.8 | 29.7 | 23.8 | 22.4 | **16.8** | 29.5 |
| MAE tS fur ($\downarrow$) | 23.7 | 23.1 | 18.1 | 17.3 | **11.6** | 23.5 |
| MAE tS obs ($\downarrow$) | 45.9 | 40.7 | 38.6 | 36.0 | **25.7** | 40.5 |
| mMAPE fr ($\downarrow$) | 29.6 | 39.2 | 26.7 | **25.9** | 39.3 | 37.5 |

latter of which is comparable to our LSTM implementation. The results of the comparisons are presented in Table 4. We observed that our proposed variants outperform the SOTA in general. The full set of results are presented in Tables 11, 12, 13, and 14.

## 5 CONCLUSIONS

This work proposes a novel regularization technique that leverages the Hottel Zone method to make deep neural networks *physics-aware* for improved furnace temperature profile prediction. Our approach is effective across various network architectures, including Multi-Layer Perceptrons (MLPs), Long Short-Term Memory (LSTM) networks, Kolmogorov-Arnold Networks (KANs) and Extended LSTM (xLSTM), as evidenced on datasets based on real-world furnace configurations with varying set points. In Sections A.9 and A.10, we respectively discuss further real-life applications of our work, along with limitations of our work and future research directions.

ACKNOWLEDGMENTS

The authors wish to acknowledge

ETHICS STATEMENT

There are no ethical concerns related to our work.

REPRODUCIBILITY STATEMENT

Sections A.4, A.6, A.8.2, and A.8.3 respectively aim at ensuring reproducibility at the following four levels: 1. Architectural and training details (e.g. number of epochs, hyper-parameters used, etc), 2. PyTorch-styled code for understanding of the implementation, 3. Algorithmic methodology used to generate dataset for ML model training, and 4. Exact data set creations and splits used for training and evaluation, with details.

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

# A APPENDIX

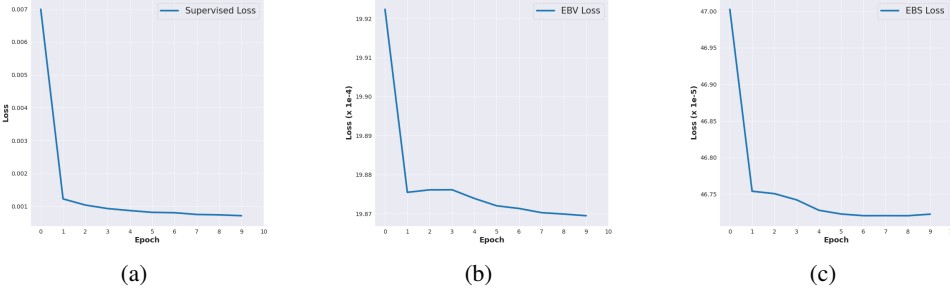

Figure 4: Convergence of PBMLP in training, considering: a) Supervised, b) EBV, and c) EBS terms.

## A.1 MOTIVATION OF OUR WORK

Yuen & Takara (1997) in their study, have proved the elegance and superiority of the zone method over contemporary counterparts to model the physical phenomenon in high-temperature processes. In our work, we use the zone method towards a real-world application for the Foundation Industries (FIs), applied to reheating furnaces, due to the close and natural association/ relation of the zone-method

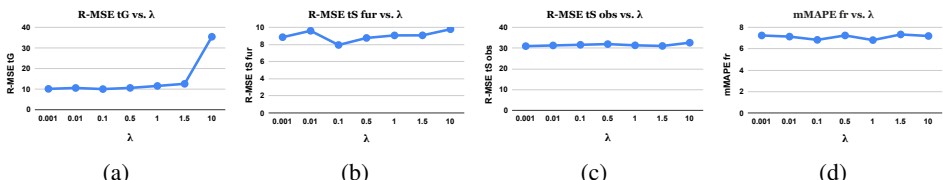

Figure 5: Performance metrics against varying $\lambda_{ebv} = \lambda_{ebs} = \lambda$ in PBMLP.

with the latter. Foundation Industries (FIs) constitute glass, metals, cement, ceramics, bulk chemicals, paper, steel, etc. and provide crucial, foundational materials for a diverse set of economically relevant industries: automobiles, machinery, construction, household appliances, chemicals, etc. FIs are heavy revenue and employment drivers, for instance, FIs in the United Kingdom (UK) economy are worth £52B (EPSRC report), employ 0.25 million people, and comprise over 7000 businesses (IOM3 report). The rapid acceleration in urbanization and industrialization over the decades has also led to improved building design and construction techniques. Great emphasis has been gradually placed on efficient heat generation, distribution, reduction, and optimized material usage.

However, despite their economic significance, as depicted by the above statistics, the FIs leverage energy-intensive methods. This makes FIs major industrial polluters and the largest consumers of natural resources across the globe. For example, in the UK, they produce 28 million tonnes of materials per year, and generate 10% of the entire UK's $CO_2$ emissions (EPSRC report; IOM3 report). Similarly, in China, the steel industry accounted for 15% of the total energy consumption, and 15.4% of the total $CO_2$ emissions (Zhang et al., 2018; Liang et al., 2020). These numbers put a challenge for the FIs in meeting our commitment to reduce net Green-House Gas (GHG) emissions, globally.

Various approaches have been relied upon to achieve the Net-Zero trajectory in FIs (Net Zero by 2050): switching of grids to low carbon alternatives via green electricity, sustainable bio-fuel, and hydrogen sources, Carbon Capture and Storage (CCS), material reuse and recycling, etc. However, among all transformation enablers, a more proactive way to address the current challenges would be to tackle the core issue of process efficiency, via digitization, computer-integrated manufacturing, and control systems. Areas of impact by digitization could be reducing plant downtime, material and energy savings, resource efficiency, and industrial symbiosis, to name a few. Various computer-aided studies have already been conducted in notable industrial scenarios. The NSG Group's Pilkington UK Limited explored a sensor-driven Machine Learning (ML) model for product quality variation prediction (up to 72h), to reduce $CO_2$ emission by 30% till 2030 (IOM3 report). Similar studies on service-oriented enterprise solutions for the steel industry have also been done recently in China (Qin et al., 2022).

In this work, we tackle the key challenge of accurate and real-time temperature prediction in reheating furnaces, which are the energy-intensive bottlenecks common across the FIs. To give a perspective to the reader on why this is important, considering any process industry, such as the steel industry, one can observe that at the core, lies the process of conversion of materials (e.g., iron) into final products. This is done using a series of unit processes (Yu et al., 2007). The production process involves key steps such as dressing, sintering, smelting, casting, rolling, etc. A nice illustration of the different stages and processes in the steel industry can be found in Qin et al. (2022). The equipment in such process industries operates in high-intensity environments (e.g., high temperature), and has bottleneck components such as reheating furnaces, which require complex restart processes post-failure. This causes additional labor costs and energy consumption. Thus, for sustainable manufacturing, it is important to monitor the operating status of the furnaces via the furnace temperature profile.

A few studies (Hu et al., 2019) have shown promise in achieving notable fuel consumption reduction by reducing the overall heating time by even as less as 13 minutes while employing alternate combustion fuels. A key area of improvement for furnace operating status monitoring lies in leveraging efficient computational temperature control mechanisms within them. This is because energy consumption per kilogram of $CO_2$ could be reduced by a reduction in overall heating time.

As existing computational surrogate models have predictive capability bottlenecks, DL approaches can be used as suitable alternatives for real-time prediction. However, as only a handful of sensors/ thermo-couples could be physically placed within real-world furnaces (and that too at specific furnace

walls), the challenge of obtaining good-quality real-world data at scale to train DL models in such scenarios remains infeasible. To alleviate this, we identify the classical Hottel's zone method (Hottel & Cohen, 1958; Hottel & Saforim, 1967) which provides an elegant, iterative way to computationally model the temperature profile within a furnace, requiring only a few initial entities which are easily measurable. However, straightforward utilization of the same is not suitable for real-time deployment and prediction, due to computational expensiveness. For this reason, we propose that we generate an offline data set using the zone method, consisting of input-output pairs to train and evaluate ML models. We will provide a detailed description of the data generation methodology using the zone method.

### A.1.1 COMPUTATIONAL MODELS

Available computational surrogate models based on Computational Fluid Dynamics (CFD) (Wehinger, 2019; De Beer et al., 2017), Discrete Element Method (DEM) (Emady et al., 2016), CFD-DEM hybrids (Oschmann & Kruggel-Emden, 2018), Two Fluid Models (TFM) (Marti et al., 2015), etc. incur expensive and time-consuming data acquisition, design, optimization, and high inference times. To break through the predictive capability bottlenecks of these surrogate models, DL approaches can be suitable candidates for real-time prediction, owing to their accuracy and inherently faster inference times (often only in the order of milliseconds).

### A.1.2 DISCUSSION ON COMPUTATIONAL ASPECTS

In general, PINNs/ PCNNs and accurate simulators (e.g., CFD models) are two different approaches to solving a physical problem. In terms of computational efficiency, they cannot be compared at the same level. While PCNNs could take milliseconds for inference, accurate simulators have difficulty even achieving real-time simulation. Thus, PCNNs have the potential to be integrated directly into a control system for real-time control. This is because PCNNs are a type of approaches that encode the governing equations of the problem into the network training, whereas, accurate simulators are based on numerical methods that discretize the problem domain and solve the equations on a mesh, which can be time-consuming, and challenging to generate for complex geometries or moving boundaries (such as the furnace studied in our work).

Generally speaking, the zone method is faster and simpler to implement than the CFD method. For example, even with a consumer-level PC, to simulate a 341-min real reheating process, the zone model only takes 5 mins, but CFD models often take several days, if not weeks, to provide *useful* results (Hu et al., 2016). Therefore, in this study, we utilize the zone model to generate training data for PCNNs. In future studies, the trained PCNNs will be integrated directly into furnace control systems. For our study, typically, generating 1500 timesteps of data for a single furnace using the zone method took about 2 hours, including the time for setting different configurations.

However, talking about the absolute time of a CFD case simulation itself depends on many factors, such as mesh density, sub-model selection, step size settings, and computer hardware configuration. Specific to our case, using the same configuration of PC, CFD simulation of the steady-state operating conditions of each setting takes about 5 hours. So the total time taken is 5 hours multiplied by the number of simulated working conditions. For the simulation of unsteady operating conditions, CFD is currently very difficult to implement, and some simplifications must be made. The specific time consumption depends on the duration of the simulated unsteady process. For the real process of 341 min for the case we studied, CFD would take at least 5 days (vs, 5 min of the zone method). As for the neural-network based implementations, for ML-based inference on a Apple M2 Max 32GB, our PCNN takes roughly 0.5s for inferring the entire furnace profile for a single time step instance, given the input variables as discussed.

### A.1.3 COMPUTATIONAL EFFICIENCY (TRAINING AND TESTING TIME) BETWEEN METHODS WITH AND WITHOUT ENERGY-BALANCE BASED PHYSICS-REGULARIZATION

The training time per mini-batch/iteration increases by up to 10x for smaller batch sizes when compared to the vanilla variant without Energy-Balance (EB) regularization. This increase is primarily due to the various matrix multiplications involving the DFA/TEA terms with higher-order matrices, particularly from the surface zones that comprise the regularization terms. However, when considering absolute run times, the increase is minimal; for example, the runtime per mini-batch is

approximately 76.11 seconds/iteration. We could reduce this further by using larger batch sizes to fully leverage GPU capabilities, although the performance gains would be marginal. In contrast, the simpler vanilla variants have a runtime of about 7.48 seconds/iteration.

During inference, the time remains the same for both variants, as the regularization terms are only required during training for the Physics-Based (PB) variants, with no changes in the architectures.

## A.2 DETAILS OF RELATED WORK

While the research conducted in this work is at nascent stage, we believe it could pave way for further developments from an ML perspective, to solve a real-world application problem with value in terms of environmental sustainability. Our work, for an applied physical sciences reader, could inspire how ML and DL could be used to address a niche domain scenario. At the same time, for an ML audience, we believe that our work showcases a novel way to integrate physics based constraints into a neural network, especially using the zone method. Arguably, there exists a plethora of works related to PINNs, however, using PINNs to incorporate the zone method based regularizers as in our work, is a novel contribution to the community. The motivation to leverage the zone method also comes from the fact that it provides an elegant (and superior) way, as studied by Yuen & Takara (1997), to model the physical phenomenon in high-temperature processes inside reheating furnaces.

In this section, we exhaustively present a set of relevant approaches with which our work can be loosely associated with. Specifically, we categorize them into two major classes: i) nonlinear dynamic systems, radiative heat transfer and view factor modeling, and, ii) modeling in reheating furnaces. We also talk about PINNs, and how our method is unique with respect to the existing literature.

**(Category 1) Nonlinear dynamic systems, radiative heat transfer and view factor modeling:** Our work at its heart is based on the zone method, which in turn relies on notions of radiative heat transfer and view factor modeling (or interchangeably, exchange area calculation). Describing the behavior of a furnace state involves combustion models, control loops, set point calculations, and fuel flux control in zones. It also involves linearization and model order reduction for state estimation and state-space control. The inherent complexity makes the modeling a nonlinear dynamic system.

While there is no exact similarity, our work shares some common philosophies with few earlier works. For instance, Ebrahimi et al. (2013) discuss the modeling of radiative heat transfer using simplified exchange area calculation. Radiative heat transfer in high-temperature thermal plasmas has been studied by Melot et al. (2011) while comparing two models. A nonlinear dynamic simulation and control based method has been studied by Hu et al. (2018). A classical work based on genetic algorithm for nonlinear dynamic systems (Li, 2005) is also present, which, instead of a data-driven approach, leverages a pre-defined set of mathematical functions.

Within this category, some approaches have also employed neural networks. In Yuen (2009), a network was trained for simulating non-gray radiative heat transfer effect in 3D gas-particle mixtures. Some approaches have used networks for view factor modeling with DEM-based simulations (Tausendschön & Radl, 2021), and some have addressed the near-field heat transfer or close regime (García-Esteban et al., 2021).

**(Category 2) Modeling in reheating furnaces:** We now discuss methods dealing with some form of prediction or optimization in reheating furnaces. Classically, Kim & Huh (2000) discussed a method to predict transient slab temperatures in a walking-beam furnace for rolling of steel slabs. Kim (2007) proposed a model for analyzing transient slab heating in a direct-fired walking beam furnace. Jang et al. (2010) investigated the slab heating characteristics with the formation and growth of scale. Tang et al. (2017) studied slab heating for process optimization. A distributed model predictive control approach was proposed in Nguyen et al. (2014). Few multi-objective optimization methods were discussed in Hu et al. (2017); Ban et al. (2023). A fuel supplies scheme based approach was proposed in Li et al. (2023). Other related works involved multi-mode model predictive control approach for steel billets (Zanoli et al., 2023), and a hybrid model for billet tapping temperature prediction (Yu et al., 2022).

Some neural network based approaches in this category studied transfer learning (Zhai & Zhou, 2020; Zhai et al., 2023), digital twin modeling (Halme Ståhlberg, 2021), and steel slab temperature prediction (de Souza Lima et al., 2023). Liao et al. (2009) discussed an integrated hybrid-PSO and fuzzy-NN decoupling based solution. Other works have studied aspects related to time-series

modeling (Hwang et al., 2019; Chen et al., 2022), and multivariate linear-regression in steel rolling (Bao et al., 2023).

**PINNs:** The methods mentioned above discuss alternatives aimed at modeling either exchange factors with radiative heat transfer, or specific slab temperature predictions in reheating furnaces. However, they do not explicitly address physics-based prior incorporation within their optimization frameworks, especially for the neural network variants. To this end, we now discuss a few relevant works in the body of literature on PINNs. For a detailed review on PINNs in general, we refer the interested reader to the papers by Raissi et al. (2019); Karniadakis et al. (2021). It should be noted that PINNs are a broad category of approaches, and the literature is vast. Here, we discuss those methods which relate to certain aspects of thermal modeling.

Drgoňa et al. (2021) proposed a physics-constrained method to model multi-zone building thermal dynamics. A multi-loss consistency optimization PINN (Shen et al., 2023) was proposed for large-scale aluminium alloy workpieces. Other approaches focus on prototype heat transfer problems and power electronics applications Cai et al. (2021), minimum film boiling temperature (Kim et al., 2022), critical heat flux (Zhao et al., 2020), solving direct and inverse heat conduction problems of materials (He et al., 2021), lifelong learning in district heating systems (Boca de Giuli, 2023), PINN and point clouds for flat plate solar collector (Han et al., 2023), residential building MPC (Bünning et al., 2022), hybrid ML and PINN for Process Control and Optimization (Park, 2022), reinforcement learning for data center cooling control (Wang et al., 2023), flexibility identification in evaporative cooling (Lahariya et al., 2022), and fast full-field temperature prediction of indoor environment (Jing et al., 2023).

**Uniqueness of our work within existing literature:** While we have observed a number of loosely related methods as discussed above, upon a clear look at them, we can conclude the following:

1. **Comparison with category 1 methods:** Among the approaches focusing on view factor modeling with radiative transfer, the area of interest is often simplified. The modeling covers select few exchange areas. The methods are also geometry-specific. Our approach on the other hand seeks a generic, geometry-agnostic modeling that covers the entire set of exchange areas. The exchange areas can be intuitively perceived as those interfaces from where radiation can transfer, between a pair of zones (surface/gas). A background on exchange areas is provided in the proposed work section.
   The ones involving neural networks, often employ feed-forward Multi-Layer Perceptron (MLP) models with few hidden layers. As showcased in our experiments, a simple MLP trained to regress the outputs given certain inputs may not generalize well to unseen distributions, due to lack of explicit understanding of the underlying physics. On the other hand, we empirically showcase that our proposed PCNN performs better than such a baseline MLP. Within a single PCNN framework, our method can also cover other architectures such as LSTMs, KANs, xLSTMs etc.

2. **Comparison with category 2 methods:** Both non-neural and neural-network based methods presented in this category, as observed, focus on predicting temperatures only in certain regions of a furnace, often, the slab temperature profiling. Our work, on the other hand aims at achieving a complete furnace temperature profiling, ranging from the gas zones, to both types of surface zones: furnace walls as well as the slab/obstacle surfaces. Our training data set is obtained based on the iterative zone method, and is more holistic in nature as compared to the discussed methods. This makes an apple-to-apple comparison difficult with other methods as they deal with different problem setups. Furthermore, the neural methods in this category are not trained to be physics aware.

3. **Comparison with PINNs:** It should be noted that any PINN approach is driven by the priors corresponding to the underlying physical phenomenon. As we did not find PINN methods addressing zone method based modeling, we could claim our PCNN variant to be novel in nature, especially, in this studied problem setup. Essentially, casting the temperature prediction task in reheating furnaces as in our work, and modeling via explicit physics-constrained regularizers (based on zone method) as done in our work, is a first of its kind. It is a simple paradigm, and could be used to build further sophisticated developments. At the same time, it simply requires input-output pairs (as shown later) to train the underlying ML/PCNN model, and makes no geometry-specific assumptions of the furnace. The data creation method discussed in our method is holistic, covers all possible exchange areas, and thus, is unique in nature itself.

## A.3 PERFORMANCE METRICS

For a data set containing $N$ samples: $\mathcal{X} = \{(\boldsymbol{x}^{(i)}, \boldsymbol{y}^{(i)})\}_{i=1}^{N}$, we make use of the following standard regression performance evaluation metrics:

1. **Root Mean Squared Error (RMSE)**, defined as:

$$RMSE = \sqrt{\frac{\sum_{i=1}^{N}(\boldsymbol{y}^{(i)} - f_\theta(\boldsymbol{x}^{(i)}))^2}{N}} \tag{13}$$

2. **Mean Absolute Error (MAE)**, defined as:

$$MAE = \frac{\sum_{i=1}^{N}\left|\boldsymbol{y}^{(i)} - f_\theta(\boldsymbol{x}^{(i)})\right|}{N} \tag{14}$$

Mean Absolute Percentage Error (MAPE) is unsuitable for firing rate prediction due to potential division by zero. We use a modified MAPE (mMAPE) with a small epsilon ($\epsilon = 0.05$) added to the denominator:

$$mMAPE = \frac{1}{N}\sum_{t=1}^{N}\left|\frac{f_t - \hat{f}_t}{f_t + \epsilon}\right| \tag{15}$$

Here, $f_t$ is the actual firing rate, and $\hat{f}_t$ is the predicted value.

We evaluate model performance for each entity (gas zone temperatures, tG; furnace surface temperatures, tS fur; obstacle surface temperatures, tS obs; firing rates, fr) separately as: RMSE tG, RMSE tS fur, RMSE tS obs, MAE tG, MAE tS fur, MAE tS obs, and mMAPE fr. Performance metrics (RMSE, MAE, mMAPE) are computed using corresponding predictions from the model ($f_\theta(\boldsymbol{x}^{(i)})$) and ground truth values from the data ($\boldsymbol{y}^{(i)}$). Results are presented for the test split (standard practice). mMAPE is evaluated only for the firing rates. RMSE, MAE and mMAPE range in $[0, \infty]$ with lower values indicating better performance ($\downarrow$) as shown in the tables.

## A.4 TRAINING DETAILS AND MODEL ARCHITECTURES

We train our PBMLP for 10 epochs using PyTorch (early stopping to avoid over-fitting), and report results with the final checkpoint. For the EB equations, we perform the same normalization for enthalpy, flux, and temperatures, as in the final neural network output as discussed earlier. We found a learning rate of 0.001 with Adam optimizer and batch size of 64 to be optimal, along with ReLU non-linearity.

We pick the [50,100,200] configuration for hidden layers, i.e., 3 hidden layers, with 50, 100, and 200 neurons respectively. We use $\lambda_{ebv} = \lambda_{ebs} = 0.1$. In general, a value lesser than 1 is observed to be better, otherwise, the model focuses less on the regression task. Following are values of other variables: $|G| = 24$, $|S| = 178$ (76 furnace surface zones and 102 obstacle surface zones), $N_g = 6$, and Stefan-Boltzmann constant=5.6687e-08. Unless otherwise stated, this is the setting we use to report any results for our method, for example, while comparing with other methods. Please note that the MLP baseline has exactly the same training configuration as the PBMLP except that it does not use the physics regularizers.

We provide details about the LSTM variants used. The LSTM variant has a single LSTM layer with 50 hidden nodes, followed by FC layer-1 with 50 input nodes and 100 output nodes, FC layer-2 with 100 input nodes and 200 output nodes. Both FC layer-1 and FC layer-2 have ReLU non-linearity. Lastly, there is a final FC layer with sigmoid nonlinearity that maps to the number of output features as in the data set. The DLSTM variant has three stacked LSTM layers, each with 100 hidden nodes, followed by a final FC layer with sigmoid nonlinearity. As we can see, we have kept the total number of layers in LSTM and DSLTM comparable to that of the baseline MLP.

For the xLSTM implementation, we follow a similar architeture as the DLSTM model. Similar to the DLSTM we place a LSTM layer that maps the input to 100 hidden nodes. However, after that, instead of stacking two more LSTM layers, we place a single xLSTM block stack (as mentioned in the official repository https://github.com/NX-AI/xlstm). After the xLSTM block, the remaining layers are similar to that of the DLSTM. Within the xLSTM block stack, the sLSTM block has 4 heads,

conv1d_kernel_size=4, and, the mLSTM block has conv1d_kernel_size=4, qkv_proj_blocksize=4, and 4 heads. Overall, xLSTM block has context length of 1, 7 blocks, and embedding dimension of 100.

For KAN, we follow the implementation suggestions as in https://github.com/KindXiaoming/pykan and use a single hidden layer with one neuron. Interestingly, the KAN despite being simpler than the MLP baseline, is not only easier to train, but also outperforms the MLP, as evidenced in many contemporary works. Broadly speaking, the training specific hyperparameters across all the compared models are the same (e.g., number of epochs, optimizer, batch size, learning rate, etc). The only difference comes from their respective architectures. For a similar architecture, the additional difference for the physics based variants lie in terms of usage of the additional regularization terms. Table 5 summarizes the details.

Table 5: Architectural and training details across different studied models

| Model | Architecture | Layer-specific information |
|---|---|---|
| MLP | 3 hidden layers (50, 100, 200 neurons)+ Final FC layer (no. of outputs) | - |
| LSTM | 1 LSTM layer (50 hidden nodes) + 2 FC layers (FC-1 and FC-2) + Final FC layer (no. of outputs) | FC-1: 50-100, FC-2: 100-200 |
| DLSTM | 3 stacked LSTM layers (100 hidden nodes each) + Final FC layer (no. of outputs) | - |
| xLSTM | 1 LSTM layer (100 hidden nodes) + 1 xLSTM block + Final FC layer (no. of outputs) | xLSTM block: context length = 1, #blocks =7, embedding dim = 100 sLSTM block:#heads=4, conv1d_kernel_size=4 mLSTM block: #heads=4, conv1d_kernel_size=4, qkv_proj_blocksize=4 |
| KAN | 1 hidden layer (1 neuron)+ Final FC layer (no. of outputs) | - |
| PB-variants | Same as corresponding base architecture, but additionally use physics-based regularizers with $\lambda_{ebv} = \lambda_{ebs} = 0.1$ | |
| | Common Hyperparameters: 10 epochs, Adam optimizer, lr=0.001, batch size=64 | |

Table 6: All results (Normal Type 1 Datasets)

| Dataset | N1-1 | | | | 925_1220_1250_750 | | | | | |
|---|---|---|---|---|---|---|---|---|---|---|
| Metric/ Method | MLP | PBMLP | LSTM | PBLSTM | DLSTM | PBDLSTM | KAN | PBKAN | xLSTM | PBxLSTM |
| RMSE tG (↓) | 136.4 | 55.3 | 15.6 | 43.3 | 28.4 | 16.1 | 40.7 | **12.6** | 39.6 | 13.7 |
| RMSE tS fur (↓) | 139.2 | 39.8 | 7.1 | 39.3 | 13.8 | **6.3** | 34.4 | 9.7 | 38.3 | 10.6 |
| RMSE tS obs (↓) | 124.8 | 64.9 | 43.7 | 73.8 | 54.2 | 52.6 | 54.2 | 21.2 | 63.9 | 22.8 |
| MAE tG (↓) | 108.6 | 51.0 | 11.1 | 39.5 | 20.7 | 10.9 | 38.8 | 10.2 | 37.5 | 11.7 |
| MAE tS fur (↓) | 115.7 | 39.2 | 6.0 | 38.1 | 12.2 | **5.1** | 34.1 | 9.1 | 37.8 | 10.0 |
| MAE tS obs (↓) | 100.2 | 54.8 | 19.5 | 58.1 | 32.1 | 22.1 | 50.1 | 18.1 | 59.3 | 18.7 |
| mMAPE fr (↓) | 232.9 | 70.7 | 25.6 | 26.5 | **21.9** | 23.7 | 51.1 | 40.7 | 22.1 | 27.6 |
| Dataset | N1-2 | | | | 965_1220_1250_750 | | | | | |
| Metric/ Method | MLP | PBMLP | LSTM | PBLSTM | DLSTM | PBDLSTM | KAN | PBKAN | xLSTM | PBxLSTM |
| RMSE tG (↓) | 113.4 | 35.6 | 33.0 | 26.7 | 117.1 | 32.4 | 24.3 | **22.6** | 130.6 | 29.3 |
| RMSE tS fur (↓) | 116.4 | 22.4 | 25.6 | **11.7** | 114.4 | 24.9 | 15.2 | 14.6 | 119.1 | 20.4 |
| RMSE tS obs (↓) | 106.9 | 43.4 | 61.1 | 66.5 | 109.3 | 67.4 | 35.1 | **33.6** | 139.8 | 45.4 |
| MAE tG (↓) | 89.5 | 28.2 | 27.4 | **16.9** | 100.9 | 27.2 | 21.4 | 19.9 | 129.1 | 26.8 |
| MAE tS fur (↓) | 96.2 | 17.8 | 21.5 | **9.9** | 101.1 | 20.1 | 14.3 | 13.8 | 118.6 | 19.5 |
| MAE tS obs (↓) | 79.9 | 29.6 | 39.4 | 31.4 | 86.9 | 44.4 | 29.8 | **29.3** | 136.3 | 39.8 |
| mMAPE fr (↓) | 176.6 | 58.5 | 29.5 | **23.5** | 201.0 | 26.2 | 44.2 | 32.6 | 200.8 | 27.8 |
| Dataset | N1-3 | | | | 995_1220_1250_750 | | | | | |
| Metric/ Method | MLP | PBMLP | LSTM | PBLSTM | DLSTM | PBDLSTM | KAN | PBKAN | xLSTM | PBxLSTM |
| RMSE tG (↓) | 31.1 | 30.5 | 39.3 | 39.2 | 100.0 | 35.7 | 23.1 | **20.9** | 114.9 | 30.1 |
| RMSE tS fur (↓) | 22.1 | 24.3 | **8.0** | 16.5 | 97.0 | 25.8 | 18.4 | 17.1 | 104.3 | 23.1 |
| RMSE tS obs (↓) | 54.4 | 47.8 | 69.0 | 77.4 | 97.2 | 60.5 | 27.7 | **26.4** | 124.2 | 35.1 |
| MAE tG (↓) | 23.0 | 23.8 | 25.3 | 29.1 | 87.0 | 29.4 | 20.9 | **18.4** | 113.6 | 27.9 |
| MAE tS fur (↓) | 16.8 | 20.8 | **6.4** | 14.6 | 85.8 | 22.4 | 17.7 | 16.4 | 104.1 | 22.4 |
| MAE tS obs (↓) | 31.4 | 29.4 | 36.6 | 46.5 | 73.1 | 32.7 | 24.0 | **22.5** | 120.7 | 30.4 |
| mMAPE fr (↓) | 32.0 | 28.1 | **25.8** | 26.9 | 128.7 | 29.4 | 33.0 | 27.7 | 127.7 | 31.7 |

A.5 FULL SET OF RESULTS ON THE 11 DATASETS

In Tables 6, 7, 8, 9 we report the performances of the compared approaches across all the 11 datasets. We noticed that not only the PB variants obtain a better performance throughout, they are also more stable across different datasets as indicated by their standard deviations (Table 10). On the other hand, the performances of the vanilla networks were not stable across different datasets.

However, we also noted that Physics-Based (PB) variants perform *slightly worse* than the vanilla methods in certain datasets. This because we did not tune hyperparameters for each configuration, but rather aimed to obtain average performance across configurations. While there may be potential for further improvements at the configuration level, our primary goal was to assess the generalizability of our approach. In real-world scenarios, variability is to be expected. It is possible that, for certain

Table 7: All results (Normal Type 2 Datasets)

| Dataset | N2-1 | | | | | | | | | |
|---|---|---|---|---|---|---|---|---|---|---|
| | | | | | 955_1190_1250_750 | | | | | |
| Metric/ Method | MLP | PBMLP | LSTM | PBLSTM | DLSTM | PBDLSTM | KAN | PBKAN | xLSTM | PBxLSTM |
| RMSE tG ($\downarrow$) | 121.1 | 45.4 | 36.8 | 37.0 | 123.4 | 28.3 | 29.5 | **18.0** | 95.5 | 33.0 |
| RMSE tS fur ($\downarrow$) | 123.8 | 27.6 | 29.5 | 28.9 | 120.5 | 18.7 | 20.7 | **8.8** | 80.6 | 24.9 |
| RMSE tS obs ($\downarrow$) | 113.1 | 52.4 | 65.6 | 63.3 | 114.5 | 51.9 | 41.0 | **27.2** | 90.7 | 51.7 |
| MAE tG ($\downarrow$) | 96.9 | 38.8 | 31.3 | 31.4 | 106.9 | 19.7 | 26.2 | **15.4** | 93.5 | 30.3 |
| MAE tS fur ($\downarrow$) | 103.6 | 24.8 | 26.7 | 25.5 | 106.4 | 16.5 | 19.8 | **7.7** | 80.1 | 24.1 |
| MAE tS obs ($\downarrow$) | 87.4 | 39.9 | 46.2 | 44.2 | 92.2 | **21.9** | 35.9 | 22.9 | 86.6 | 46.5 |
| mMAPE fr ($\downarrow$) | 187.6 | 67.8 | 28.4 | 29.8 | 210.6 | **24.9** | 43.7 | 34.2 | 212.3 | 26.2 |
| Dataset | N2-2 | | | | | | | | | |
| | | | | | 955_1230_1250_750 | | | | | |
| Metric/ Method | MLP | PBMLP | LSTM | PBLSTM | DLSTM | PBDLSTM | KAN | PBKAN | xLSTM | PBxLSTM |
| RMSE tG ($\downarrow$) | 116.1 | 39.2 | 34.3 | 34.6 | 122.5 | 33.3 | 27.6 | **18.0** | 135.5 | 31.0 |
| RMSE tS fur ($\downarrow$) | 118.6 | 24.3 | 28.4 | 27.9 | 119.9 | 27.3 | 19.6 | **9.7** | 123.9 | 23.9 |
| RMSE tS obs ($\downarrow$) | 108.7 | 45.2 | 64.0 | 61.7 | 113.6 | 70.7 | 39.6 | **29.0** | 144.8 | 50.2 |
| MAE tG ($\downarrow$) | 91.1 | 32.9 | 29.5 | 29.7 | 105.4 | 28.9 | 24.7 | **15.5** | 134.0 | 28.7 |
| MAE tS fur ($\downarrow$) | 96.7 | 20.8 | 25.8 | 24.6 | 105.8 | 23.9 | 18.8 | **8.8** | 123.3 | 23.2 |
| MAE tS obs ($\downarrow$) | 82.8 | 32.5 | 44.4 | 42.5 | 91.2 | 49.6 | 34.4 | **24.6** | 141.3 | 44.9 |
| mMAPE fr ($\downarrow$) | 187.1 | 66.7 | 28.4 | 30.0 | 220.4 | **25.6** | 46.8 | 35.0 | 220.6 | 26.7 |

Table 8: All results (Normal Type 3 Datasets)

| Dataset | N3-1 | | | | | | | | | |
|---|---|---|---|---|---|---|---|---|---|---|
| | | | | | 955_1220_1250_750 | | | | | |
| Metric/ Method | MLP | PBMLP | LSTM | PBLSTM | DLSTM | PBDLSTM | KAN | PBKAN | xLSTM | PBxLSTM |
| RMSE tG ($\downarrow$) | 119.5 | 42.9 | 34.4 | 34.7 | 122.7 | 33.3 | 27.6 | **18.0** | 135.5 | 31.0 |
| RMSE tS fur ($\downarrow$) | 122.5 | 24.1 | 28.5 | 27.9 | 120.1 | 27.4 | 19.6 | **9.7** | 123.9 | 23.9 |
| RMSE tS obs ($\downarrow$) | 111.3 | 45.5 | 64.1 | 61.9 | 113.7 | 70.7 | 39.6 | **28.8** | 144.8 | 50.2 |
| MAE tG ($\downarrow$) | 94.6 | 36.6 | 29.6 | 29.7 | 105.5 | 28.9 | 24.7 | **15.5** | 134.1 | 28.7 |
| MAE tS fur ($\downarrow$) | 101.5 | 20.3 | 25.8 | 24.7 | 105.9 | 24.0 | 18.8 | **8.7** | 123.3 | 23.2 |
| MAE tS obs ($\downarrow$) | 85.1 | 33.3 | 44.4 | 42.6 | 91.3 | 49.6 | 34.4 | **24.5** | 141.3 | 44.9 |
| mMAPE fr ($\downarrow$) | 194.2 | 88.0 | 28.4 | 30.0 | 220.4 | **25.6** | 46.8 | 35.0 | 220.6 | 26.6 |
| Dataset | N3-2 | | | | | | | | | |
| | | | | | 955_1220_1280_750 | | | | | |
| Metric/ Method | MLP | PBMLP | LSTM | PBLSTM | DLSTM | PBDLSTM | KAN | PBKAN | xLSTM | PBxLSTM |
| RMSE tG ($\downarrow$) | 23.8 | 17.9 | 19.5 | 19.5 | 17.3 | 18.1 | 14.9 | **14.5** | 16.4 | 15.9 |
| RMSE tS fur ($\downarrow$) | 11.2 | 7.8 | 12.0 | 11.2 | 9.6 | 10.5 | **6.8** | 7.3 | 9.4 | 9.2 |
| RMSE tS obs ($\downarrow$) | 57.6 | 41.6 | 54.5 | 52.0 | 61.9 | 61.6 | **26.0** | 26.7 | 33.9 | 34.8 |
| MAE tG ($\downarrow$) | 17.0 | 11.8 | 14.7 | 14.6 | 13.1 | 13.7 | 12.0 | **11.7** | 14.1 | 13.7 |
| MAE tS fur ($\downarrow$) | 9.6 | 6.8 | 10.7 | 9.6 | 8.0 | 8.6 | **6.0** | 6.6 | 8.6 | 8.3 |
| MAE tS obs ($\downarrow$) | 31.5 | **20.1** | 27.7 | 26.2 | 32.3 | 32.5 | 20.9 | 21.5 | 27.7 | 28.6 |
| mMAPE fr ($\downarrow$) | 37.5 | 41.9 | 25.2 | 27.2 | **22.1** | 22.9 | 51.2 | 50.6 | 21.5 | 22.9 |
| Dataset | N3-3 | | | | | | | | | |
| | | | | | 955_1220_1300_750 | | | | | |
| Metric/ Method | MLP | PBMLP | LSTM | PBLSTM | DLSTM | PBDLSTM | KAN | PBKAN | xLSTM | PBxLSTM |
| RMSE tG ($\downarrow$) | 18.2 | 15.6 | 15.6 | **15.5** | 15.6 | **15.5** | 17.5 | 19.0 | 12.5 | 11.5 |
| RMSE tS fur ($\downarrow$) | 7.5 | 8.7 | 7.7 | **7.0** | 7.6 | 7.7 | 11.2 | 13.7 | 5.9 | 6.0 |
| RMSE tS obs ($\downarrow$) | 52.4 | 47.2 | 51.2 | 48.3 | 58.7 | 58.4 | **27.6** | 29.2 | 28.1 | 28.7 |
| MAE tG ($\downarrow$) | 11.0 | 11.7 | 10.2 | 10.2 | 11.3 | 11.2 | 15.2 | 17.1 | 10.7 | **10.0** |
| MAE tS fur ($\downarrow$) | 6.0 | 7.1 | 6.0 | 5.4 | 6.4 | 6.4 | 10.6 | 13.0 | 5.4 | **5.3** |
| MAE tS obs ($\downarrow$) | 23.4 | 24.4 | 22.2 | **21.1** | 26.1 | 26.3 | 23.2 | 24.8 | 22.5 | 22.9 |
| mMAPE fr ($\downarrow$) | 40.5 | 38.7 | 27.9 | 30.5 | **22.9** | 24.9 | 60.2 | 62.3 | 21.3 | 24.0 |

configurations, the underlying physics is better captured by a stronger vanilla architecture (e.g., LSTM vs. MLP). If the vanilla model is effectively learning and generalizing, the explicit regularization may yield minimal gains. However, we do not consider this a case of PB variants performing worse than vanilla methods; rather, their performance metrics are comparable.

Conversely, it is important to note that PB variants generally outperform vanilla variants by significant multiplicative factors in performance metrics.

The performances of the proposed Physics-Based (PB) approaches across all the 11 datasets are also compared against the following SOTA methods: i) MLRVPST (Bao et al. (2023)) and ii) PTDL-LSTM (de Souza Lima et al. (2023)), the results of which are presented in Tables 11, 12, 13, and 14. We notice that our proposed variants outperform the SOTA consistently in general.

A.6   PSEUDO-CODES FOR OUR TRAINING FRAMEWORK

In Algorithm 2, we outline the key steps required in training our physics-constrained framework. The training involves a typical mini-batch based optimization, where each instance in a mini-batch contains the various entities obtained from one row/time step of the data set. The entities are present in their respective columns. The columns for the constant terms (e.g., $(\dot{Q}_{conv})_i$, $(\dot{Q}_{fuel,net})_i$, $(\dot{Q}_a)_i$, $A_i(\dot{q}_{conv})_i$ and $\dot{Q}_{s,i}$) will have the values repeated across all the corresponding rows to create a dataloader.

Table 9: All results (Normal Type 4 Datasets)

| Dataset | N4-1 | | | | 955_1220_1250_705 | | | | | |
|---|---|---|---|---|---|---|---|---|---|---|
| Metric/ Method | MLP | PBMLP | LSTM | PBLSTM | DLSTM | PBDLSTM | KAN | PBKAN | xLSTM | PBxLSTM |
| RMSE tG ($\downarrow$) | 117.4 | 39.3 | 110.8 | 34.2 | 29.6 | 31.5 | 27.1 | **17.3** | 93.3 | 92.9 |
| RMSE tS fur ($\downarrow$) | 121.9 | 32.9 | 98.2 | 30.2 | 19.7 | 26.3 | 20.6 | **8.6** | 80.1 | 79.1 |
| RMSE tS obs ($\downarrow$) | 115.7 | 64.3 | 126.2 | 67.3 | 48.7 | 53.4 | 47.0 | **23.1** | 94.6 | 94.7 |
| MAE tG ($\downarrow$) | 94.2 | 35.3 | 90.0 | 30.3 | 22.0 | 24.2 | 28.7 | **14.4** | 91.8 | 91.2 |
| MAE tS fur ($\downarrow$) | 102.0 | 31.6 | 78.3 | 27.2 | 17.9 | 20.5 | 22.0 | **7.7** | 79.7 | 78.5 |
| MAE tS obs ($\downarrow$) | 91.5 | 51.6 | 92.1 | 50.7 | 21.4 | 30.2 | 55.9 | **19.4** | 90.6 | 90.7 |
| mMAPE fr ($\downarrow$) | 123.0 | 19.9 | 141.7 | 21.6 | 22.3 | 28.0 | 22.4 | **17.2** | 139.9 | 141.0 |
| Dataset | N4-2 | | | | 955_1220_1250_765 | | | | | |
| Metric/ Method | MLP | PBMLP | LSTM | PBLSTM | DLSTM | PBDLSTM | KAN | PBKAN | xLSTM | PBxLSTM |
| RMSE tG ($\downarrow$) | 38.7 | 36.1 | 34.2 | 24.2 | 121.9 | 32.4 | 27.3 | **18.0** | 135.5 | 30.5 |
| RMSE tS fur ($\downarrow$) | 27.0 | 23.2 | 27.9 | 13.4 | 119.3 | 26.6 | 19.3 | **10.2** | 123.8 | 23.5 |
| RMSE tS obs ($\downarrow$) | 63.9 | 44.4 | 61.9 | 65.7 | 111.4 | 69.2 | 37.3 | **31.2** | 142.6 | 47.9 |
| MAE tG ($\downarrow$) | 32.7 | 29.5 | 29.2 | **15.1** | 104.5 | 27.9 | 24.5 | 15.6 | 134.2 | 28.3 |
| MAE tS fur ($\downarrow$) | 24.3 | 19.5 | 25.1 | 12.2 | 105.1 | 23.2 | 18.5 | **9.4** | 123.2 | 22.8 |
| MAE tS obs ($\downarrow$) | 45.7 | 30.0 | 41.8 | 29.5 | 88.9 | 47.5 | 31.9 | **26.8** | 139.2 | 42.4 |
| mMAPE fr ($\downarrow$) | 42.9 | 59.7 | 30.2 | **23.3** | 229.6 | 25.7 | 49.8 | 37.0 | 230.2 | 27.6 |
| Dataset | N4-3 | | | | 955_1220_1250_810 | | | | | |
| Metric/ Method | MLP | PBMLP | LSTM | PBLSTM | DLSTM | PBDLSTM | KAN | PBKAN | xLSTM | PBxLSTM |
| RMSE tG ($\downarrow$) | 35.5 | 28.0 | 35.3 | **25.2** | 120.3 | 30.2 | 27.0 | 33.4 | 27.6 | 29.4 |
| RMSE tS fur ($\downarrow$) | 21.8 | 19.3 | 25.5 | **8.7** | 117.1 | 23.8 | 18.1 | 27.4 | 20.9 | 21.9 |
| RMSE tS obs ($\downarrow$) | **46.1** | 48.0 | 53.2 | 67.5 | 105.7 | 62.8 | 31.7 | 51.8 | 40.6 | 42.1 |
| MAE tG ($\downarrow$) | 25.5 | 20.3 | 29.0 | **15.4** | 102.6 | 24.7 | 24.4 | 31.3 | 24.7 | 27.1 |
| MAE tS fur ($\downarrow$) | 16.4 | 14.7 | 21.8 | **7.3** | 103.0 | 19.5 | 17.5 | 26.5 | 19.4 | 21.1 |
| MAE tS obs ($\downarrow$) | 28.8 | **27.1** | 33.2 | 32.1 | 82.4 | 38.9 | 26.5 | 47.9 | 34.4 | 36.1 |
| mMAPE fr ($\downarrow$) | 57.5 | 50.0 | 40.3 | **24.6** | 259.6 | 28.2 | 61.0 | 60.0 | 28.0 | 30.6 |

Table 10: All results (standard deviations)

| Dataset | STDEV | | | | | | | | | |
|---|---|---|---|---|---|---|---|---|---|---|
| Metric/ Method | MLP | PBMLP | LSTM | PBLSTM | DLSTM | PBDLSTM | KAN | PBKAN | xLSTM | PBxLSTM |
| RMSE tG ($\downarrow$) | 48.2 | 11.6 | 25.9 | 8.7 | 48.8 | 7.5 | 6.7 | 5.4 | 51.1 | 21.8 |
| RMSE tS fur ($\downarrow$) | 55.8 | 9.2 | 25.4 | 10.9 | 52.4 | 8.3 | 6.8 | 5.8 | 48.3 | 19.4 |
| RMSE tS obs ($\downarrow$) | 31.2 | 8.0 | 21.6 | 8.4 | 27.6 | 7.1 | 8.7 | 8.1 | 47.2 | 18.8 |
| MAE tG ($\downarrow$) | 39.3 | 11.8 | 21.5 | 9.5 | 43.3 | 7.3 | 7.0 | 5.5 | 51.5 | 21.9 |
| MAE tS fur ($\downarrow$) | 46.4 | 9.5 | 20.2 | 10.4 | 46.2 | 7.2 | 7.0 | 5.8 | 48.5 | 19.5 |
| MAE tS obs ($\downarrow$) | 30.0 | 10.8 | 19.3 | 11.3 | 30.2 | 10.6 | 11.0 | 8.0 | 48.2 | 19.1 |
| mMAPE fr ($\downarrow$) | 78.3 | 20.2 | 34.2 | 3.1 | 99.7 | 2.0 | 11.1 | 13.5 | 91.6 | 34.4 |

Table 11: All results against SOTA (Normal Type 1 Datasets)

| Dataset | N1-1 | | | | | |
|---|---|---|---|---|---|---|
| Metric/ Method | MLRVPST | PTDL-LSTM | PBLSTM | PBDLSTM | PBKAN | PBxLSTM |
| RMSE tG ($\downarrow$) | 45.4 | 15.6 | 43.3 | 16.1 | **12.6** | 13.7 |
| RMSE tS fur ($\downarrow$) | 41.0 | 7.1 | 39.3 | **6.3** | 9.7 | 10.6 |
| RMSE tS obs ($\downarrow$) | 68.6 | 43.7 | 73.8 | 52.6 | **21.2** | 22.8 |
| MAE tG ($\downarrow$) | 43.2 | 11.1 | 39.5 | 10.9 | **10.2** | 11.7 |
| MAE tS fur ($\downarrow$) | 40.5 | 6.0 | 38.1 | **5.1** | 9.1 | 10.0 |
| MAE tS obs ($\downarrow$) | 64.2 | 19.5 | 58.1 | 22.1 | **18.1** | 18.7 |
| mMAPE fr ($\downarrow$) | 28.4 | 25.6 | 26.5 | **23.7** | 40.7 | 27.6 |
| Dataset | N1-2 | | | | | |
| Metric/ Method | MLRVPST | PTDL-LSTM | PBLSTM | PBDLSTM | PBKAN | PBxLSTM |
| RMSE tG ($\downarrow$) | 30.7 | 33.0 | 26.7 | 32.4 | **22.6** | 29.3 |
| RMSE tS fur ($\downarrow$) | 22.1 | 25.6 | **11.7** | 24.9 | 14.6 | 20.4 |
| RMSE tS obs ($\downarrow$) | 48.8 | 61.1 | 66.5 | 67.4 | **33.6** | 45.4 |
| MAE tG ($\downarrow$) | 28.1 | 27.4 | **16.9** | 27.2 | 19.9 | 26.8 |
| MAE tS fur ($\downarrow$) | 21.3 | 21.5 | **9.9** | 20.1 | 13.8 | 19.5 |
| MAE tS obs ($\downarrow$) | 43.2 | 39.4 | 31.4 | 44.4 | **29.3** | 39.8 |
| mMAPE fr ($\downarrow$) | 31.8 | 29.5 | **23.5** | 26.2 | 32.6 | 27.8 |
| Dataset | N1-3 | | | | | |
| Metric/ Method | MLRVPST | PTDL-LSTM | PBLSTM | PBDLSTM | PBKAN | PBxLSTM |
| RMSE tG ($\downarrow$) | 27.8 | 39.3 | 39.2 | 35.7 | **20.9** | 30.1 |
| RMSE tS fur ($\downarrow$) | 20.6 | **8.0** | 16.5 | 25.8 | 17.1 | 23.1 |
| RMSE tS obs ($\downarrow$) | 36.7 | 69.0 | 77.4 | 60.5 | **26.4** | 35.1 |
| MAE tG ($\downarrow$) | 25.1 | 25.3 | 29.1 | 29.4 | **18.4** | 27.9 |
| MAE tS fur ($\downarrow$) | 19.4 | **6.4** | 14.6 | 22.4 | 16.4 | 22.4 |
| MAE tS obs ($\downarrow$) | 31.5 | 36.6 | 46.5 | 32.7 | **22.5** | 30.4 |
| mMAPE fr ($\downarrow$) | 32.3 | **25.8** | 26.9 | 29.4 | 27.7 | 31.7 |

As observed in Algorithm 2, X_train_batch and y_train_batch correspond to $\boldsymbol{x}^{(i)}$ and $\boldsymbol{y}^{(i)}$ in $\mathcal{X}$, and are used to compute tr_loss_regtmps representing $\mathcal{L}_{sup}$ in eq(12). tr_loss_ebv and tr_loss_ebs respectively correspond to $\mathcal{L}_{ebv}$ and $\mathcal{L}_{ebs}$ in eq(12). The collection of the $T_g$ terms for being associated with the computational graph for back-propagation by virtue of use in eq(8), is done by y_train_pred[:,:n_gas_zones].

Table 12: All results against SOTA (Normal Type 2 Datasets)

| Dataset | | | N2-1 | | | |
|---|---|---|---|---|---|---|
| Metric/ Method | MLRVPST | PTDL-LSTM | PBLSTM | PBDLSTM | PBKAN | PBxLSTM |
| RMSE tG ($\downarrow$) | 35.7 | 36.8 | 37.0 | 28.3 | **18.0** | 33.0 |
| RMSE tS fur ($\downarrow$) | 27.8 | 29.5 | 28.9 | 18.7 | **8.8** | 24.9 |
| RMSE tS obs ($\downarrow$) | 55.5 | 65.6 | 63.3 | 51.9 | **27.2** | 51.7 |
| MAE tG ($\downarrow$) | 32.8 | 31.3 | 31.4 | 19.7 | **15.4** | 30.3 |
| MAE tS fur ($\downarrow$) | 27.0 | 26.7 | 25.5 | 16.5 | **7.7** | 24.1 |
| MAE tS obs ($\downarrow$) | 50.5 | 46.2 | 44.2 | **21.9** | 22.9 | 46.5 |
| mMAPE fr ($\downarrow$) | 30.6 | 28.4 | 29.8 | **24.9** | 34.2 | 26.2 |

| Dataset | | | N2-2 | | | |
|---|---|---|---|---|---|---|
| Metric/ Method | MLRVPST | PTDL-LSTM | PBLSTM | PBDLSTM | PBKAN | PBxLSTM |
| RMSE tG ($\downarrow$) | 33.4 | 34.3 | 34.6 | 33.3 | **18.0** | 31.0 |
| RMSE tS fur ($\downarrow$) | 26.3 | 28.4 | 27.9 | 27.3 | **9.7** | 23.9 |
| RMSE tS obs ($\downarrow$) | 53.5 | 64.0 | 61.7 | 70.7 | **29.0** | 50.2 |
| MAE tG ($\downarrow$) | 30.8 | 29.5 | 29.7 | 28.9 | **15.5** | 28.7 |
| MAE tS fur ($\downarrow$) | 25.5 | 25.8 | 24.6 | 23.9 | **8.8** | 23.2 |
| MAE tS obs ($\downarrow$) | 48.3 | 44.4 | 42.5 | 49.6 | **24.6** | 44.9 |
| mMAPE fr ($\downarrow$) | 31.6 | 28.4 | 30.0 | **25.6** | 35.0 | **26.7** |

Table 13: All results against SOTA (Normal Type 3 Datasets)

| Dataset | | | N3-1 | | | |
|---|---|---|---|---|---|---|
| Metric/ Method | MLRVPST | PTDL-LSTM | PBLSTM | PBDLSTM | PBKAN | PBxLSTM |
| RMSE tG ($\downarrow$) | 33.5 | 34.4 | 34.7 | 33.3 | **18.0** | 31.0 |
| RMSE tS fur ($\downarrow$) | 26.5 | 28.5 | 27.9 | 27.4 | **9.7** | 23.9 |
| RMSE tS obs ($\downarrow$) | 53.7 | 64.1 | 61.9 | 70.7 | **28.8** | 50.2 |
| MAE tG ($\downarrow$) | 31.0 | 29.6 | 29.7 | 28.9 | **15.5** | 28.7 |
| MAE tS fur ($\downarrow$) | 25.7 | 25.8 | 24.7 | 24.0 | **8.7** | 23.2 |
| MAE tS obs ($\downarrow$) | 48.5 | 44.4 | 42.6 | 49.6 | **24.5** | 44.9 |
| mMAPE fr ($\downarrow$) | 31.4 | 28.4 | 30.0 | **25.6** | 35.0 | 26.6 |

| Dataset | | | N3-2 | | | |
|---|---|---|---|---|---|---|
| Metric/ Method | MLRVPST | PTDL-LSTM | PBLSTM | PBDLSTM | PBKAN | PBxLSTM |
| RMSE tG ($\downarrow$) | 18.0 | 19.5 | 19.5 | 18.1 | **14.5** | 15.9 |
| RMSE tS fur ($\downarrow$) | 11.4 | 12.0 | 11.2 | 10.5 | **7.3** | 9.2 |
| RMSE tS obs ($\downarrow$) | 38.1 | 54.5 | 52.0 | 61.6 | **26.7** | 34.8 |
| MAE tG ($\downarrow$) | 15.7 | 14.7 | 14.6 | 13.7 | **11.7** | 13.7 |
| MAE tS fur ($\downarrow$) | 10.7 | 10.7 | 9.6 | 8.6 | **6.6** | 8.3 |
| MAE tS obs ($\downarrow$) | 32.0 | 27.7 | 26.2 | 32.5 | **21.5** | 28.6 |
| mMAPE fr ($\downarrow$) | 27.2 | 25.2 | 27.2 | **22.9** | 50.6 | **22.9** |

| Dataset | | | N3-3 | | | |
|---|---|---|---|---|---|---|
| Metric/ Method | MLRVPST | PTDL-LSTM | PBLSTM | PBDLSTM | PBKAN | PBxLSTM |
| RMSE tG ($\downarrow$) | 14.0 | 15.6 | 15.5 | 15.5 | 19.0 | **11.5** |
| RMSE tS fur ($\downarrow$) | 8.2 | 7.7 | 7.0 | 7.7 | 13.7 | **6.0** |
| RMSE tS obs ($\downarrow$) | 32.5 | 51.2 | 48.3 | 58.4 | 29.2 | **28.7** |
| MAE tG ($\downarrow$) | 11.3 | 10.2 | 10.2 | 11.2 | 17.1 | **10.0** |
| MAE tS fur ($\downarrow$) | 7.3 | 6.0 | 5.4 | 6.4 | 13.0 | **5.3** |
| MAE tS obs ($\downarrow$) | 26.3 | 22.2 | **21.1** | 26.3 | 24.8 | 22.9 |
| mMAPE fr ($\downarrow$) | 28.9 | 27.9 | 30.5 | 24.9 | 62.3 | **24.0** |

Similar role towards back-propagation via $T_s$ terms in eq(9) is taken care of by `y_train_pred[:,n_gas_zones:n_gas_zones+n_fur_surf_zones+n_obs_surf_zones]`.

`get_pb_ebv_pred()` computes $\boldsymbol{v}_g$ in eq(10) for each instance (corresponding to a time-step of zone method) present in a mini-batch of the variables obtained from the already created data set. In doing so, each of the $|G|$ elements of $\boldsymbol{v}_g$ are computed using eq(8) and the corresponding/relevant auxiliary variables from the data. `sgarr_plus_hg_tensor_batch` collects mini-batch terms using relevant terms like $\boldsymbol{s}_{(g)arr}, \boldsymbol{h}_g$ in eq(10) towards $\boldsymbol{v}_g$. The relevant DFA terms are collected in tensor `dfa_GG_tensor_batch`. Similarly, we make use of `get_pb_ebs_pred()`, `dfa_SS_tensor_batch`, `gsarr_plus_hs_tensor_batch` for computing $\boldsymbol{v}_s$ in eq(10) and using eq(9). Having obtained the dataset, it only involves sampling mini-batches via appropriate helper functions in any Deep Learning framework (e.g., PyTorch). In Algorithms 3-4, we provide a few helper functions which can be useful to further understand the computation of some of the tensors involved in the training loop described in Algorithm 2.

## A.7 IN-DEPTH SENSITIVITY ANALYSIS OF PBMLP

We evaluated PBMLP's sensitivity to hyperparameters (loss terms, hidden layers, batch size, activation functions) using shuffled test data from all furnace configurations. To establish an upper bound on

Table 14: All results against SOTA (Normal Type 4 Datasets)

| Dataset | N4-1 | | | | | |
|---|---|---|---|---|---|---|
| Metric/ Method | MLRVPST | PTDL-LSTM | PBLSTM | PBDLSTM | PBKAN | PBxLSTM |
| RMSE tG ($\downarrow$) | 36.2 | 110.8 | 34.2 | 31.5 | **17.3** | 92.9 |
| RMSE tS fur ($\downarrow$) | 30.9 | 98.2 | 30.2 | 26.3 | **8.6** | 79.1 |
| RMSE tS obs ($\downarrow$) | 62.5 | 126.2 | 67.3 | 53.4 | **23.1** | 94.7 |
| MAE tG ($\downarrow$) | 33.9 | 90.0 | 30.3 | 24.2 | **14.4** | 91.2 |
| MAE tS fur ($\downarrow$) | 30.4 | 78.3 | 27.2 | 20.5 | **7.7** | 78.5 |
| MAE tS obs ($\downarrow$) | 57.9 | 92.1 | 50.7 | 30.2 | **19.4** | 90.7 |
| mMAPE fr ($\downarrow$) | 20.2 | 141.7 | 21.6 | 28.0 | **17.2** | 141.0 |
| Dataset | N4-2 | | | | | |
| Metric/ Method | MLRVPST | PTDL-LSTM | PBLSTM | PBDLSTM | PBKAN | PBxLSTM |
| RMSE tG ($\downarrow$) | 32.2 | 34.2 | 24.2 | 32.4 | **18.0** | 30.5 |
| RMSE tS fur ($\downarrow$) | 25.0 | 27.9 | 13.4 | 26.6 | **10.2** | 23.5 |
| RMSE tS obs ($\downarrow$) | 50.8 | 61.9 | 65.7 | 69.2 | **31.2** | 47.9 |
| MAE tG ($\downarrow$) | 29.7 | 29.2 | **15.1** | 27.9 | 15.6 | 28.3 |
| MAE tS fur ($\downarrow$) | 24.2 | 25.1 | 12.2 | 23.2 | **9.4** | 22.8 |
| MAE tS obs ($\downarrow$) | 45.3 | 41.8 | 29.5 | 47.5 | **26.8** | 42.4 |
| mMAPE fr ($\downarrow$) | 32.6 | 30.2 | **23.3** | 25.7 | 37.0 | 27.6 |
| Dataset | N4-3 | | | | | |
| Metric/ Method | MLRVPST | PTDL-LSTM | PBLSTM | PBDLSTM | PBKAN | PBxLSTM |
| RMSE tG ($\downarrow$) | 36.8 | 35.3 | **25.2** | 30.2 | 33.4 | 29.4 |
| RMSE tS fur ($\downarrow$) | 29.4 | 25.5 | **8.7** | 23.8 | 27.4 | 21.9 |
| RMSE tS obs ($\downarrow$) | 61.2 | 53.2 | 67.5 | 62.8 | 51.8 | 42.1 |
| MAE tG ($\downarrow$) | 34.9 | 29.0 | **15.4** | 24.7 | 31.3 | 27.1 |
| MAE tS fur ($\downarrow$) | 28.9 | 21.8 | **7.3** | 19.5 | 26.5 | 21.1 |
| MAE tS obs ($\downarrow$) | 57.2 | 33.2 | **32.1** | 38.9 | 47.9 | 36.1 |
| mMAPE fr ($\downarrow$) | 30.7 | 40.3 | **24.6** | 28.2 | 60.0 | 30.6 |

performance, we employed teacher forcing during evaluation (providing ground truth values from previous time steps as inputs). This explains the improved metrics compared to auto-regressive real-world like inference from earlier tables.

We observed good convergence of PBMLP (Fig 4), with the default setting mentioned in Appendix A.4. Table 15 shows performance with different hidden layer configurations, with [50, 100, 200] providing competitive results. Here, [100] denotes one hidden layer with 100 neurons, [50, 100] denotes two hidden layers with 50, and 100 neurons respectively, and so on. The maximum values for each row (corresponding to a metric) are shown in bold. In Table 16, we vary the batch size in our method. We found a batch size of 64 to provide an optimal performance for our experiments. In our exploration of activation functions, ReLU, SiLU, and Mish exhibited similar performance, with ReLU proving more robust across batch sizes (Table 18).

We also examined all possible combinations of the regularizer weights $\lambda_{ebv}$ and $\lambda_{ebs}$. Table 17 highlights extreme cases where one regularizer is set to zero while the other is at a higher value, i.e., keeping only the EBV term by setting $\lambda_{ebv} = 0.1$ and $\lambda_{ebs} = 0$, and only the EBS term by setting $\lambda_{ebv} = 0$ and $\lambda_{ebs} = 0.1$. We found that performance is better while using both regularizers together rather than in isolation.

However, we found that excessively high values for the regularizers can compete with the regression loss terms, a common issue noted in PINN literature. Specifically, when $\lambda_{ebs}$ is set too high, it can significantly degrade performance due to the larger number of surface zones typically present in a furnace overpowering the loss function. Based on these observations and to avoid unnecessary complexity with varying values (e.g., 0.1, 0.3, etc), which resulted in minimal performance differences, we opted for a single value of $\lambda_{ebv}$ and $\lambda_{ebs}$ for the sensitivity analysis for both regularizers. This decision simplifies our design while ensuring optimal learning rate adjustments are considered. The results are presented in Figure 5 where we observe a stable performance across values except a drop in R-MSE tG at $\lambda_{ebs} = 10$ as mentioned.

A.8  DATA DETAILS: FROM FURNACE TO ML MODEL TRAINING AND EVALUATION

We now discuss the data set details of our benchmarking. Prior to discussing the data used for ML model training and evaluation, we provide the reader a brief flavor on the physical understanding of a real-world furnace, along with its operation.

**Algorithm 2** PyTorch-styled pseudo-code for training loop of our framework

```
1
2  ### TRAINING ###
3  criterion = nn.MSELoss()
4  optimizer = optim.Adam(model.parameters(), lr=LEARNING_RATE)
5  for e in tqdm(range(1, EPOCHS+1)):
6      model.train()
7      for (batch_idx, sample_batched) in enumerate(train_loader_EBVS):
8          #sample_batched[0]:data, sample_batched[1]:labels, sample_batched[2]:auxvars
9          X_train_batch = sample_batched[0].to(device)
10         y_train_batch = sample_batched[1].to(device)
11         auxvars_dict_batch = sample_batched[2]
12
13         dfa_GG_tensor_batch = auxvars_dict_batch['dfa_GG_tensor'].to(device)
14         sgarr_plus_hg_tensor_batch = auxvars_dict_batch['sgarr_plus_hg'].to(device)
15         dfa_SS_tensor_batch = auxvars_dict_batch['dfa_SS_tensor'].to(device)
16         gsarr_plus_hs_tensor_batch = auxvars_dict_batch['gsarr_plus_hs'].to(device)
17
18         optimizer.zero_grad()
19
20         y_train_pred = model(X_train_batch)
21         tr_loss_regtmps = criterion(y_train_pred, y_train_batch)
22
23         ## EBV terms
24         pb_ebv_pred = get_pb_ebv_pred(
25             sgarr_plus_hg_tensor_batch, dfa_GG_tensor_batch,
26             y_train_pred[:,:n_gas_zones]
27             )
28         pb_ebv_actual = torch.zeros(pb_ebv_pred.size()).to(device)
29
30         ## EBS terms
31         pb_ebs_pred = get_pb_ebs_pred(
32             gsarr_plus_hs_tensor_batch, dfa_SS_tensor_batch,
33             y_train_pred[:,n_gas_zones:n_gas_zones+n_fur_surf_zones+n_obs_surf_zones]
34             )
35         pb_ebs_actual = torch.zeros(pb_ebs_pred.size()).to(device)
36
37         tr_loss_ebv = criterion(pb_ebv_pred, pb_ebv_actual) / y_train_pred.size(0)
38         tr_loss_ebs = criterion(pb_ebs_pred, pb_ebs_actual) / y_train_pred.size(0)
39
40         batch_loss=tr_loss_regtmps+lambda_ebv*tr_loss_ebv+lambda_ebs*tr_loss_ebs
41         batch_loss.backward()
42         optimizer.step()
```

Table 15: Performance of PBMLP (ReLU) variant of our method against varying hidden layer configurations .

| Metric/ Hidden layer configuration | [100] | [50,100] | [50,100, 200] | [50,100, 200,200] | [50,100, 200,200, 205,205] |
|---|---|---|---|---|---|
| RMSE tG ($\downarrow$) | 11.64 | 17.25 | **10.04** | 10.84 | 14.27 |
| RMSE tS fur ($\downarrow$) | 10.05 | 15.23 | 7.95 | **7.83** | 12.46 |
| RMSE tS obs ($\downarrow$) | 34.82 | 37.62 | **31.64** | 33.57 | 36.42 |
| mMAPE fr ($\downarrow$) | 8.76 | 9.15 | **6.84** | 8.06 | 7.51 |

Table 16: Performance of the proposed PBMLP variant using different batch sizes .

| Metric | PBMLP ReLU bsz=32 | PBMLP ReLU bsz=64 | PBMLP ReLU bsz=128 |
|---|---|---|---|
| RMSE tG ($\downarrow$) | 12.70 | **10.04** | 10.73 |
| RMSE tS fur ($\downarrow$) | 9.14 | **7.95** | 9.69 |
| RMSE tS obs ($\downarrow$) | 39.75 | **31.64** | 31.79 |
| mMAPE fr ($\downarrow$) | **5.24** | 6.84 | 8.29 |

Table 17: Effect of individual regularizer terms in PBMLP .

| Metric | EBV only | EBS only | PBMLP |
|---|---|---|---|
| RMSE tG ($\downarrow$) | 11.85 | 11.66 | **10.04** |
| RMSE tS fur ($\downarrow$) | 10.36 | 11.07 | **7.95** |
| RMSE tS obs ($\downarrow$) | 32.46 | 32.04 | **31.64** |
| mMAPE fr ($\downarrow$) | **6.42** | 7.53 | 6.84 |

Table 18: Performance of PBMLP using different activation functions in the underlying network.

| Metric | PBMLP ReLU | PBMLP GeLU | PBMLP SiLU | PBMLP Hardswish | PBMLP Mish |
|---|---|---|---|---|---|
| RMSE tG ($\downarrow$) | **10.04** | 13.57 | 10.07 | 15.26 | 10.16 |
| RMSE tS fur ($\downarrow$) | 7.95 | 8.86 | 8.02 | 14.02 | **7.71** |
| RMSE tS obs ($\downarrow$) | 31.64 | 39.65 | 31.64 | 36.23 | **31.63** |
| mMAPE fr ($\downarrow$) | 6.84 | **5.88** | 6.23 | 7.03 | 6.33 |

### A.8.1 BACKGROUND ON FURNACE OPERATION

For experimentation, we consider a real-world, walking beam top-fired furnace in Swerim (former Swerea MEFOS), Sweden, which has been studied by Hu et al. Hu et al. (2019). Figure 6 illustrates the furnace, which can be conceptually subdivided into several zones along both its length and height, such as dark, control, and soaking, which represent regions with distinct temperatures. It has varying

**Algorithm 3** PyTorch-styled pseudo-code for helper functions in our framework

```
1
2  ### HELPER FUNCTIONS ###
3
4  # For EBV
5  dfa_GG_tensor_all = get_dfa_AB_tensor_all(
6      tea_GG, get_torch_float(X_tG_gaszone_prev).to(device)
7      )
8  sgarr_plus_hg_all = get_sgarr_plus_hg_all(
9      get_torch_float(X_hg).to(device), tea_GS,
10     torch.hstack(( get_torch_float(X_tS_furnace_prev),
11         get_torch_float(X_tS_obstacle_prev) )).to(device)
12     )
13
14 def get_pb_ebv_pred_instance(sgarr_plus_hg_tensor, dfa_GG_tensor, tG_single_pred):
15     ## computes \mathbf{v}_g vector for one time step
16
17 def get_pb_ebv_pred(sgarr_plus_hg_tensor_batch, dfa_GG_tensor_batch, y_train_pred_only_tG):
18     ## calls get_pb_ebv_pred_instance for all instances in the batch
19
20 # For EBS
21 dfa_SS_tensor_all = get_dfa_AB_tensor_all(
22     tea_SS, get_torch_float(np.hstack(
23         [X_tS_furnace_prev, X_tS_obstacle_prev]
24         )).to(device))
25 gsarr_plus_hs_all = get_gsarr_plus_hs_all(
26     get_torch_float(X_hs).to(device), tea_SG,
27     get_torch_float(X_tG_gaszone_prev).to(device)
28     )
29
30 def get_pb_ebs_pred_instance(gsarr_plus_hs_tensor, dfa_SS_tensor, tS_single_pred):
31     ## computes \mathbf{v}_s vector for one time step
32
33 def get_pb_ebs_pred(gsarr_plus_hs_tensor_batch, dfa_SS_tensor_batch, y_train_pred_only_tS):
34     ## calls get_pb_ebs_pred_instance for all instances in the batch
```

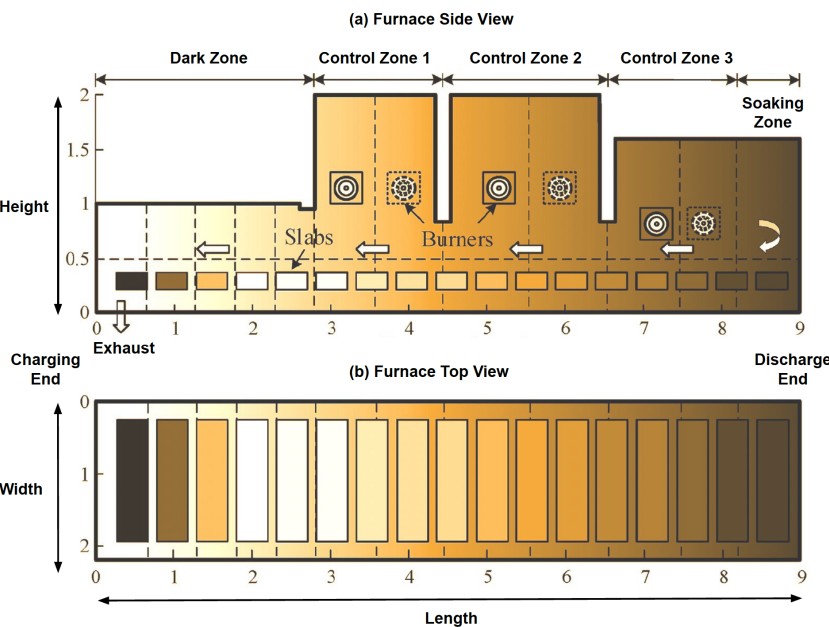

Figure 6: Illustration of the real-world furnace in Swerim, Sweden, and its subdivision as different zones Hu et al. (2019). Figure is best viewed in color. The temperature increases towards the discharge end (at the right), as indicated by a darker shade. The slabs are heated while moving from the left to the right.

heights for different zones but is of fixed length and width. It has a target heating temperature of 1250 °C and its production capacity is 3 tonne/hr. Reheating furnaces are used to heat intermediate steel products usually known as stock (e.g., blooms, billets, slabs).

**Algorithm 4** PyTorch-styled pseudo-code for additional helper functions in our framework

```
1
2  ### HELPER FUNCTIONS (set 2) ###
3
4  def inverse_transform_Vectorized_pt(scaledtensor, range, min_along_dims, dist):
5      range_min, range_max=range
6      origtensor = min_along_dims+dist*(scaledtensor-range_min)/(range_max - range_min)
7      return origtensor
8
9  def get_an_mat_tensor(tB_singlerow_tensor):
10     tMat_tensor=torch.tile(tB_singlerow_tensor, (Ng, 1))
11     coef_b_mat_T=coef_b_mat.T
12     for ii in range(coef_b_mat_T.shape[1]):# Taylor series loop
13         bn=coef_b_mat_T[:,[ii]]
14         bn_tensor=torch.from_numpy(bn).float().to(device)
15         if ii==0:
16             an_mat_tensor=torch.mul(torch.tile(
17                 bn_tensor, (1, tMat_tensor.size(1))),tMat_tensor**ii)
18         else:
19             an_mat_tensor+=torch.mul(torch.tile(
20                 bn_tensor, (1, tMat_tensor.size(1))),tMat_tensor**ii)
21     return an_mat_tensor
22
23 def get_pb_ebv_pred_instance(sgarr_plus_hg_tensor, dfa_GG_tensor, tG_single_pred):
24     startid_col, endid_col=0, n_gas_zones
25
26     tG_current_tensor = inverse_transform_Vectorized_pt(
27         tG_single_pred,(0,1), ytr_min_along_dims[[0], startid_col:endid_col].to(device),
28         ytr_dist[[0], startid_col:endid_col].to(device))
29
30     ggarr_tensor=torch.sum(torch.mul( dfa_GG_tensor , sbcons*torch.tile(
31         tG_current_tensor**4, (dfa_GG_tensor.size(0), 1)) ),1, keepdim=True).T
32
33     an_mat_G_tensor=get_an_mat_tensor(tG_current_tensor)
34
35     tmpmat2=sbcons*torch.mul( torch.tile(
36         Vi_current_tensor ,(an_mat_G_tensor.size(0),1) ) ,
37         torch.tile(tG_current_tensor**4, (an_mat_G_tensor.size(0), 1)) )
38     tmpmat1=torch.mul( an_mat_G_tensor , torch.tile(
39         coef_k_mat_T_tensor , (1,an_mat_G_tensor.size(1))) )
40     gleave_tensor=torch.sum(torch.mul(tmpmat1,tmpmat2),0,keepdim=True)
41
42     pb_ebv_pred_instance= torch.abs(ggarr_tensor+sgarr_plus_hg_tensor-4*gleave_tensor)
43     pb_ebv_pred_instance/=pb_ebv_pred_instance.max(dim=1, keepdim=True)[0]
44
45
46     return pb_ebv_pred_instance
```

Through a series of discrete pushes, the transport of slabs occurs within a furnace. As shown in Figure 6, a first slab at an ambient temperature is pushed from the charging end at the left side of furnace (lower temperature, shown in a lighter shade). At each push, all slabs move forward towards the discharge end at the right (higher temperature, shown in a darker shade). For a few specific regions in the furnace, the process operator pre-defines a few **set point temperatures**, which indicate the temperatures to which the slabs must be heated. The slabs once heated to the required set point temperatures, are collected at the discharge end. The movement of the slabs is controlled by the **walk-interval** (walk rate), depending on the desired throughput.

The internal combustion is controlled via **firing rates** of a few burners located in specific regions. In Figure 6, we can see that there are six burners: 2 in each of control zones 1, 2, and 3. In this particular furnace, the pair of burners in a control zone share the same firing rate values. Note that these firing rates are normalized in $[0, 1]$.

Describing the behavior of a furnace state involves combustion models, control loops, set point calculations, and fuel flux control in zones. It also involves linearization and model order reduction for state estimation and state-space control. The inherent complexity makes the modeling a nonlinear dynamic system. We provide set point temperatures, walk interval, firing rates and initial state of the furnace (indicated by temperatures of various gas and surface regions/zones in it) as inputs to this system. These inputs, along with the overall movement of the slabs within the furnace, influence the mass and energy flow throughout the furnace system. This, in turn, results in a new furnace state, characterized by a new set of temperatures.

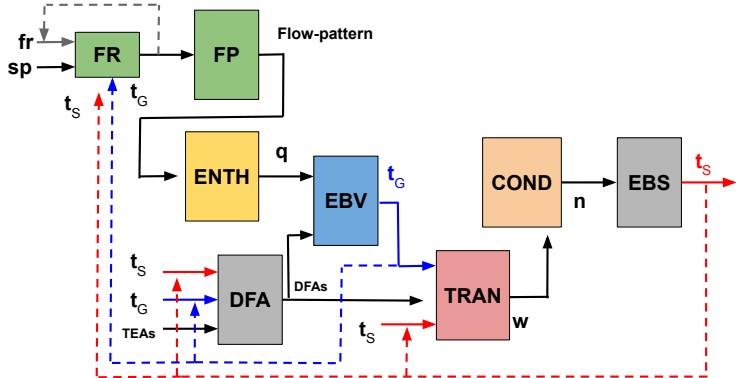

Figure 7: Illustration of flow of the data generation algorithm. The figure is best viewed in color. Dashed lines denote feedback from past time step. Blue/red/gray lines correspond for $t_G$/$t_S$/$fr$, respectively. Block Abbreviations are, FR: Firing Rate, FP: Flow-pattern, ENTH: Enthalpy, TRAN: Heat-transfer, COND: Conduction analysis, EBV/S: Energy-Balance Volume/Surface, and DFA: Directed Flux Area. Details of components present in the text.

The ideal scenario involves a computational model that can predict the next set of temperatures based on the provided inputs. This predicted state can then be compared to the desired set point temperatures. Deviations from the set points trigger adjustments in the firing rates. If a region's predicted temperature falls short of the set point, the firing rate for the corresponding burner increases. Conversely, if the predicted temperature exceeds the desired value, the firing rate is lowered. A Proportional-Integral-Derivative (PID) controller is employed to manage these adjustments in practice. This controller factors in the walk interval to ensure smooth and controlled changes in the firing rates, ultimately leading to a furnace state that aligns with the set point temperatures.

### A.8.2 PROPOSED DATA GENERATION METHODOLOGY FOR TEMPERATURE PREDICTION USING ML

As shown in Figure 6, it is possible to conceptually divide the furnace into 1, 2, and 12 sections across its width, height, and length respectively. This results in a total of 24 **volume/gas zones**, where gaseous material could reside. These zones can be visualized using the dashed vertical and horizontal lines in the figure.

Additionally, at a time step, there can be 17 slabs inside the furnace, each of which has 6 surfaces, thus, resulting in 102 slab surfaces. With prior knowledge of the 3D structure of our furnace, we computed a total of 76 furnace walls, which could be called furnace surfaces. We can respectively call the 102 slab surfaces as obstacle/ slab surface zones, and the 76 furnace walls as furnace surface zones. Collectively, the obstacle/ slab surface zones and furnace surface zones result in a total of 178 **surface zones**, which in addition to the volume zones form the basis of utilization of the Hottel's zone method.

The flow of combustion products within the furnace results in heat release. This causes radiation interchange among all possible pairs of zones: gas to gas, surface to surface, and surface to gas (and vice-versa). The dominating heat transfer mechanism in such processes is Radiative Heat Transfer (RHT), which naturally occurs among the other heat transfer mechanisms: conduction and convection. For each pair of zones, there would be an **energy balance**, i.e., the amount of energy entering a zone would equal the amount leaving it. To model the RHT, the zone method subdivides an enclosure into a finite number of isothermal volume and surface **zones**, and applies energy balance to each of them. In our case, for example, we have a total of 202 zones (178 for surfaces and 24 for volumes).

We can model the radiative exchange among any two zones by leveraging underlying governing physical equations, and *energy balances*. The zone method also employs pre-computed exchange areas (which are general forms of view factors). The main objective is to then compute unknown parameters such as temperatures (of volumes and surfaces), and heat fluxes. This could be done by solving a set of simultaneous equations. We direct the interested reader to Yuen & Takara (1997); Hu et al. (2016; 2019), for a better perspective of the zone method.

We shall design the data framework in such a way that it can easily plug in any standard ML (or DL) model for regression. For this, notice that although the various entities within the computational method depend on the geometry of the furnace, we can make a learnable model agnostic of the geometry, if we can train it by simply using data in the form of input-output pairs, and (optional) auxiliary/ intermediate variables (say, for regularization).

One simple way is to collect all relevant values from across zones corresponding to an entity in the form of a vector. For example, we could collect all gas zone temperatures within a vector, and likewise, for other entities such as surface zones, enthalpies, heat fluxes, node temperatures, etc, we could form individual vectors. This gives us the freedom to ignore the 3D structure during training as we can simply deal with vectors and their mappings, say within a neural network, or any other ML technique. Post-inference analysis or fine-grained process control could later be performed via our knowledge of which zone an attribute of the vector maps to.

In Figure 7, we present our proposed algorithmic flow mimicking the Hottel's zone method Hottel & Cohen (1958); Hottel & Saforim (1967); Yuen & Takara (1997) based computational model of Hu et al. Hu et al. (2016), for data generation aimed at training regression-based ML models. In this, notice how we represent all the relevant entities as vectors. While we shall discuss all relevant terms of the zone method in detail, during the explanation of the modeling part, we now briefly give an overview of the various stages of the zone method. Here, let $\Phi$ represents a particular block/ stage, and $\theta$ represents the applicable parameters for the underlying function (abbreviated name shown in the subscript). Following are the stages in the generation method (represented by a block in Figure 7):

1. **Firing Rates updation block ($\Phi_{\theta_{fr}}$):** Using the predicted gas ($t_G$) and surface ($t_S$) zone temperatures from a previous time step, a calibration against the setpoint temperatures provided in **sp** is performed to update the firing rates **fr** for the current time step (also denoted as $f$). In Figure 7 we use slightly abused notations of **fr** and **sp** to represent firing rates and setpoints for avoiding confusion with other notations such as *surface*.

2. **DFA block ($\Phi_{\theta_{dfa}}$):** Notice that for a time step, the inputs $t_S, t_G$ are obtained from the corresponding values obtained as outputs in the previous time step, shown respectively by dashed red and blue backward arrows. Here, $|S|$ and $|G|$ denote the total number of surface and gas zones, and, $t_S \in \mathbb{R}^{|S|}, t_G \in \mathbb{R}^{|G|}$ are vectors collecting all the surface zone and gas/volume zone temperatures respectively. Hu et al Hu et al. (2016), using an updated Monte-Carlo based Ray-Tracing (MCRT) algorithm Matthew et al. (2014), provide fixed, pre-computed Total Exchange Areas (TEAs) (forms of view factors Yuen & Takara (1997)) as inputs along with $t_S, t_G$, for computing the Radiation Exchange factors, or the Directed Flux Area (DFA) terms.

   The TEAs are denoted as: $GS \in \mathbb{R}^{|G| \times |S| \times N_g}, SS \in \mathbb{R}^{|S| \times |S| \times N_g}, GG \in \mathbb{R}^{|G| \times |G| \times N_g}$, and $SG \in \mathbb{R}^{|S| \times |G| \times N_g}$ (we can drop the third dimension for the sake of brevity). Here, $GS, SS, GG$, and $SG$ contain the pre-computed gas-surface, surface-surface, gas-gas, and surface-gas exchange areas. $\overleftarrow{GS} \in \mathbb{R}^{|G| \times |S|}, \overleftarrow{SS} \in \mathbb{R}^{|S| \times |S|}, \overleftarrow{GG} \in \mathbb{R}^{|G| \times |G|}$, and $\overleftarrow{SG} \in \mathbb{R}^{|S| \times |G|}$ are the corresponding DFA terms for $GS, SS, GG$, and $SG$ respectively ($\leftarrow$ indicates the direction of flow). Here, $N_g$ denotes the number of gases used for representing a real gas medium.

   Initially, we assume that a steady-state has been reached, and hence assign ambient temperature values to $t_S, t_G$. The parameters $\theta_{dfa}$ represent fixed correlation coefficients (as discussed in the methodology section).

3. **Flow pattern ($\Phi_{\theta_{fp}}$) and enthalpy blocks ($\Phi_{\theta_{enth}}$):** Given initial firing rates in $f \in \mathbb{R}^{|B|}$ ($|B|$ is a function of the number of burners), the block representing the function $\Phi_{\theta_{fp}}$ obtains the flow pattern $flat(F)$, which is further used by the block representing the function $\Phi_{\theta_{enth}}$ to obtain the enthalpy vector $q$.

   Note that, the flow of combustion gases within an enclosure causes mass flow into (+ve) and out (-ve) of a zone, for each inter-zone boundary plane. This flow could be pre-computed in a CPU instantly using a polynomial fitted through isothermal CFD simulations that define a range of experimental points, derived with Box–Behnken designs Ferreira et al. (2007). The flow pattern resulted is by nature a matrix $F \in \mathbb{R}^{|G| \times 12}$, but the spatial dependency among the matrix elements can be discarded for simplicity, and we can rather represent an equivalent flattened vector $flat(F) \in \mathbb{R}^{12|G|}$ obtained in row-major fashion. Note that, as already mentioned, we subdivide an enclosure into several cubes/ boxes (zones in our

**Algorithm 5** Data generation algorithm for a fixed furnace configuration

1: Initialize a steady-state furnace configuration via set points and walk interval.
2: Initialize $\mathcal{X} = \{\}$, $T > 0$ (max no. of steps).
3: Initialize $\boldsymbol{t}_G^{(0)}$, $\boldsymbol{t}_S^{(0)}$ with steady-state ambient temperatures, and $\boldsymbol{f}^{(0)}$.
4: **for** t=1 **to** $T$ **do**                                                                    ▷ t: time-step
5:      $\boldsymbol{f}^{(t)} \leftarrow \Phi_{\theta_{fr}}(\boldsymbol{f}^{(t-1)}, \text{set point temperatures}, \boldsymbol{t}_G^{(t-1)}, \boldsymbol{t}_S^{(t-1)})$
6:      $\boldsymbol{q}^{(t)} \leftarrow \Phi_{\theta_{enth}}(\Phi_{\theta_{fp}}(\boldsymbol{f}^{(t)}))$
7:      $\overleftarrow{\boldsymbol{GG}}^{(t)}, \overleftarrow{\boldsymbol{GS}}^{(t)}, \overleftarrow{\boldsymbol{SG}}^{(t)}, \overleftarrow{\boldsymbol{SS}}^{(t)} \leftarrow \Phi_{\theta_{dfa}}(\boldsymbol{t}_G^{(t-1)}, \boldsymbol{t}_S^{(t-1)}, \boldsymbol{GG}, \boldsymbol{GS}, \boldsymbol{SG}, \boldsymbol{SS})$
8:      $\boldsymbol{t}_G^{(t)} \leftarrow \Phi_{\theta_{ebv}}(\boldsymbol{q}^{(t)}, \overleftarrow{\boldsymbol{GG}}^{(t)}, \overleftarrow{\boldsymbol{GS}}^{(t)})$
9:      $\boldsymbol{w}^{(t)} \leftarrow \Phi_{\theta_{tran}}(\boldsymbol{t}_G^{(t)}, \boldsymbol{t}_S^{(t-1)}, \overleftarrow{\boldsymbol{SS}}^{(t)}, \overleftarrow{\boldsymbol{SG}}^{(t)})$
10:     $\boldsymbol{t}_S^{(t)} \leftarrow \Phi_{\theta_{ebs}}(\boldsymbol{n}^{(t)})$, where $\boldsymbol{n}^{(t)} \leftarrow \Phi_{\theta_{con}}(\boldsymbol{w}^{(t)})$
11:     $\mathcal{X}_t \leftarrow \{\boldsymbol{f}^{(t)}, \boldsymbol{F}^{(t)}, \boldsymbol{q}^{(t)}, \boldsymbol{t}_S^{(t)}, \boldsymbol{t}_G^{(t)}, \boldsymbol{w}^{(t)}, \boldsymbol{n}^{(t)}\}$
12:     $\mathcal{X} \leftarrow \mathcal{X} \cup \mathcal{X}_t$
13: **end for**
14: **return** $\mathcal{X}$

case). Since any cube has 6 surfaces, and for each surface we have two directions of flow (+ve and -ve), this results in 12 flows for each volume zone, and thus, the 12 arises in the dimensionality of $\boldsymbol{F}$.

Also, for each volume zone $i$, we would require an enthalpy transport term $(\dot{Q}_{enth})_i$. We introduce an enthalpy vector $\boldsymbol{q} \in \mathbb{R}^{|G|}$ to compactly represent these terms.

4. **Energy Balance Volume (EBV) block ($\Phi_{\theta_{ebv}}$):** We introduce a block to compute the volume zone temperatures $\boldsymbol{t}_G$ using the enthalpy vector $\boldsymbol{q}$ and the DFA terms $\overleftarrow{\boldsymbol{GG}}$ and $\overleftarrow{\boldsymbol{GS}}$.

5. **Heat transfer block ($\Phi_{\theta_{tran}}$):** Together with the volume zone temperatures $\boldsymbol{t}_G$, the obtained DFAs ($\overleftarrow{\boldsymbol{SS}}, \overleftarrow{\boldsymbol{SG}}$), and the previously obtained (or initialized) surface zone temperatures $\boldsymbol{t}_S$, we obtain the **heat transfer/ flux** to the surfaces as a variable $\boldsymbol{w}$.

6. **Conduction analysis block ($\Phi_{\theta_{con}}$):** The heat flux on each surface zone serves as a boundary condition for performing a conduction analysis, to compute the transient heat conduction through each surface. The conduction process results in the node temperatures, which we represent as a variable $\boldsymbol{n}$.

7. **Energy Balance Surface (EBS) block ($\Phi_{\theta_{ebs}}$):** The computation of **heat transfer/ flux** and surface zone temperatures are coupled together as the surface energy balance equations. Having computed the heat transfer and performing the conduction analysis, the surface zone temperatures in $\boldsymbol{t}_S$ can be updated using the node temperatures $\boldsymbol{n}$. This is a fixed function.

**The Algorithm:** Algorithm 5 presents the steps involved in the data generation method. We assume that for a steady-state furnace configuration (with fixed set points and walk interval), our data set is in the form: $\mathcal{X} = \{\mathcal{X}_t\}_{t=1}^T$, where, $\mathcal{X}_t = \{\boldsymbol{f}^{(t)}, \boldsymbol{F}^{(t)}, \boldsymbol{q}^{(t)}, \boldsymbol{t}_S^{(t)}, \boldsymbol{t}_G^{(t)}, \boldsymbol{w}^{(t)}, \boldsymbol{n}^{(t)}\}$ is the set of observed variables as described in Figure 7, for a time-step $t$. Note that the computations of flow patterns, enthalpy, and node temperatures can be treated independently from the energy balance equations.

| | timestep | firing_rates | walk_interval | setpoints | flowpattern | q_enthalpy | tG_gaszone | tS_furnace | tS_obstacle | w_flux_furnace | ... |
|---|---|---|---|---|---|---|---|---|---|---|---|
| 0 | 1000035 | [0.162, 0.9, 0.689] | 750 | [905.0, 1220.0, 1250.0] | [0.27214, 0.00037, 0.0, 0.0, 0.15124, 0.00502,... | [325971.875, 6805.781, 16632.312, 20740.859, 2... | [1238.396, 655.898, 669.693, 720.935, 783.621,... | [899.66, 696.459, 676.871, 707.375, 759.241, 8... | [282.33, 198.022, 230.603, 267.441, 244.599, 2... | [1227.219, 61.728, 44.997, 77.785, 123.674, 26... | ... |
| 1 | 1000050 | [0.176, 0.9, 0.697] | 750 | [905.0, 1220.0, 1250.0] | [0.27379, 0.00031, 0.0, 0.0, 0.15469, 0.00493,... | [331067.125, 6830.078, 16803.453, 20947.594, 2... | [1245.547, 657.297, 670.983, 722.349, 785.105,... | [900.576, 696.454, 676.84, 707.373, 759.285, 8... | [291.843, 205.389, 239.773, 277.841, 253.712, ... | [1470.822, 138.764, 84.222, 121.113, 176.747, ... | ... |
| 2 | 1000065 | [0.188, 0.9, 0.705] | 750 | [905.0, 1220.0, 1250.0] | [0.27532, 0.00027, 0.0, 0.0, 0.15768, 0.00486,... | [335621.75, 6849.953, 16960.344, 21137.922, 24... | [1252.052, 658.657, 672.223, 723.702, 786.523,... | [901.643, 696.504, 676.845, 707.41, 759.375, 8... | [301.287, 212.751, 248.861, 288.102, 262.75, 2... | [1680.182, 211.778, 121.823, 162.165, 226.299,... | ... |

Figure 8: Sample training data instances for each time step within a configuration.

Figure 8 illustrates a few sample time steps (in rows), and the corresponding entities (in columns) generated by using Algorithm 5. The full list of entities that we generate for a time step is: 'timestep', 'firing_rates', 'walk_interval', 'setpoints', 'flowpattern', 'q_enthalpy', 'tG_gaszone', 'tS_furnace', 'tS_obstacle', 'w_flux_furnace',

'w_flux_obstacle', 'nodetmp_1d_furnace', 'nodetmp_2d_obstacle'. The names of the entities are self-explanatory (e.g., 'nodetmp_1d_furnace' refers to 1D node temperatures for furnace surfaces, 'nodetmp_2d_obstacle' refers to 2D node temperatures for obstacle surfaces), where G as usual, denotes *gas zone* and S denotes *surface zone*, the latter, is further divided into *furnace* and *obstacle*.

| | tG_gaszone_prev | tS_furnace_prev | tS_obstacle_prev | firing_rates | tG_gaszone | tS_furnace | tS_obstacle | firing_rates_next |
|---|---|---|---|---|---|---|---|---|
| 0 | [1230.741, 654.484, 668.378, 719.49, 782.103, ... | [898.918, 696.524, 676.938, 707.417, 759.248, ... | [272.753, 190.658, 221.352, 256.904, 235.417, ... | [0.162, 0.9, 0.689] | [1238.396, 655.898, 669.693, 720.935, 783.621,... | [899.66, 696.459, 676.871, 707.375, 759.241, 8... | [282.33, 198.022, 230.603, 267.441, 244.599, 2... | [0.176, 0.9, 0.697] |
| 1 | [1238.396, 655.898, 669.693, 720.935, 783.621,... | [899.66, 696.459, 676.871, 707.375, 759.241, 8... | [282.33, 198.022, 230.603, 267.441, 244.599, 2... | [0.176, 0.9, 0.697] | [1245.547, 657.297, 670.983, 722.349, 785.105,... | [900.576, 696.454, 676.84, 707.373, 759.285, 8... | [291.843, 205.389, 239.773, 277.841, 253.712, ... | [0.188, 0.9, 0.705] |
| 2 | [1245.547, 657.297, 670.983, 722.349, 785.105,... | [900.576, 696.454, 676.84, 707.373, 759.285, 8... | [291.843, 205.389, 239.773, 277.841, 253.712, ... | [0.188, 0.9, 0.705] | [1252.052, 658.657, 672.223, 723.702, 786.523,... | [901.643, 696.504, 676.845, 707.41, 759.375, 8... | [301.287, 212.751, 248.861, 288.102, 262.75, 2... | [0.197, 0.9, 0.712] |
| 3 | [1252.052, 658.657, 672.223, 723.702, 786.523,... | [901.643, 696.504, 676.845, 707.41, 759.375, 8... | [301.287, 212.751, 248.861, 288.102, 262.75, 2... | [0.197, 0.9, 0.712] | [1257.793, 659.953, 673.385, 724.964, 787.842,... | [902.832, 696.606, 676.883, 707.482, 759.508, ... | [310.652, 220.1, 257.862, 298.222, 271.709, 27... | [0.209, 0.9, 0.718] |
| 4 | [1257.793, 659.953, 673.385, 724.964, 787.842,... | [902.832, 696.606, 676.883, 707.482, 759.508, ... | [310.652, 220.1, 257.862, 298.222, 271.709, 27... | [0.209, 0.9, 0.718] | [1263.848, 661.255, 674.595, 726.284, 789.244,... | [904.15, 696.761, 676.954, 707.59, 759.686, 82... | [319.959, 227.441, 266.784, 308.212, 280.599, ... | [0.218, 0.9, 0.727] |

Figure 9: Rearranged training data instances (selected columns).

Assuming that the original data is stored in a Pandas DataFrame (using a Python syntax), for each time step we also need the following entities: 'firing_rates_next', 'tG_gaszone_prev', 'tS_furnace_prev', and 'tS_obstacle_prev'. This is because, for computing the entities in a time step, we make use of the temperatures in the previous time step. At the same time, for experimental purposes, we also try to directly predict the next firing rate via ML. Thus, using Python syntax, we could perform the following:

a) `df['firing_rates_next'] = df['firing_rates'].shift(-1)` followed by `df = df.drop(df.tail(1).index)`.
b) `df['tG_gaszone_prev']=df['tG_gaszone'].shift(1)`, `df['tS_furnace_prev'] = df['tS_furnace'].shift(1)`, `df['tS_obstacle_prev'] = df['tS_obstacle'].shift(1)` followed by `df = df.drop(df.head(1).index)`.

The rearranged data can be visualized as in Figure 9 (we only showcase relevant entities here, owing to limited space). Essentially, we add a new column 'firing_rates_next' by shifting the original firing rates column a step back and then dropping the last row. Likewise, we add new columns for *prev* temperatures by shifting the original temperature columns a step forward and then dropping the first row. Please note that some additional auxiliary variables are used by the computational method of Hu et al. Hu et al. (2016), which are mostly constants, and could thus be repeated/ copied for each time step. They are: 'corrcoeff_b', 'Qconvi', 'extinctioncoeff_k', 'gasvolumes_Vi', 'QfuelQa_sum', 'surfareas_Ai', 'emissivity_epsi', 'convection_flux_qconvi'. We later leverage them in training our PCNN, with the help of regularizers.

Now we can form any data set containing $N$ samples: $\mathcal{X} = \{(\boldsymbol{x}^{(i)}, \boldsymbol{y}^{(i)})\}_{i=1}^N$ to train an off-the-shelf, standard ML/ DL model $f_\theta(.)$ with learnable parameters $\theta$, which expects an input instance $\boldsymbol{x}^{(i)}$ as vector and predicts an output vector $\boldsymbol{y}^{(i)}$, i.e., $\boldsymbol{y}^{(i)} = f_\theta(\boldsymbol{x}^{(i)})$. Here, $\boldsymbol{x}^{(i)}$ and $\boldsymbol{y}^{(i)}$ can be formed using entities from desired columns obtained from the rearranged data as shown in Figure 9. Notice how the above proposed ML training framework via our data generation in the form of simple input-output pairs lets any generic regression model learn freely without requiring 3D geometry-specific knowledge during the training. This makes our proposed framework geometry-agnostic, and hence flexible by nature to accommodate any ML method.

### A.8.3 BENCHMARKING DATA SET DETAILS FOR ML MODEL DEVELOPMENT AND EVALUATION

Algorithm 5 outlines data generation for a fixed furnace configuration (defined by set points and walk interval). Set points are desired temperatures for certain zones. We represent a configuration as: `SP1_SP2_SP3_WI`, where `SP1`, `SP2`, `SP3` and `WI` respectively denote the set point 1, set point 2, set point 3, and walk interval. Under normal conditions naturally occurring in practice, following will hold true: `SP1<SP2<SP3`. For robustness, we consider 50 configurations (based on the furnace in Fig 6) and generate corresponding *configuration datasets*, including abnormal configurations with arbitrary set points. Since each dataset has a unique configuration, their inherent data distributions differ.

From the 50 distinct datasets, we combine configurations (e.g., first, fourth, seventh) to form a consolidated training split. Similar combinations create validation and test splits with no overlap between them. This creates a test bed to evaluate model generalization across different data distributions, crucial for real-world deployment where inference data might differ from training data. Table 19 details these configurations, indicating their membership in training, validation, or test splits, within parentheses. Test datasets (e.g., N1-2, N1-3) are named based on their set point characteristics and are also shown in bold.

It should be noted that the default SP1,SP2,SP3,WI setting is kept: `955_1220_1250_750`. With this, we vary each of SP1, SP2, SP3, and WI with certain step-size. This leads to four groups/types of configurations within the Normal Behaviour Configurations shown in Table 19. The nomenclature of the test data sets is done to indicate their grouping, e.g., prefixes N1-, N2-, N3- and N4- denote whether the configuration belongs to the group with varying SP1, SP2, SP3, and WI respectively. Thus, Ni-j indicates the j-th configuration of the group i, and is used to represent a test *configuration data set*. As it can be seen, there are **11 normal test data sets** where we evaluate the ML models.

Table 19: Benchmark data details.

| Normal Behaviour Configurations (SP1<SP2<SP3) | | | |
|---|---|---|---|
| **Type 1 (Varying SP1 only)** | **Type 2 (Varying SP2 only)** | **Type 3 (Varying SP3 only)** | **Type 4 (Varying WI only)** |
| 905_1220_1250_750 (Training)
915_1220_1250_750 (Val)
**925_1220_1250_750 (N1-1)**
935_1220_1250_750 (Training)
945_1220_1250_750 (Val)
**965_1220_1250_750 (N1-2)**
975_1220_1250_750 (Training)
985_1220_1250_750 (Val)
**995_1220_1250_750 (N1-3)** | 955_1170_1250_750 (Training)
955_1180_1250_750 (Val)
**955_1190_1250_750 (N2-1)**
955_1200_1250_750 (Training)
955_1210_1250_750 (Val)
**955_1230_1250_750 (N2-2)**
955_1240_1250_750 (Training) | 955_1220_1230_750 (Training)
955_1220_1240_750 (Val)
**955_1220_1250_750 (N3-1)**
955_1220_1260_750 (Training)
955_1220_1270_750 (Val)
**955_1220_1280_750 (N3-2)**
955_1220_1290_750 (Training)
**955_1220_1300_750 (N3-3)** | 955_1220_1250_675 (Training)
955_1220_1250_690 (Val)
**955_1220_1250_705 (N4-1)**
955_1220_1250_720 (Training)
955_1220_1250_735 (Val)
**955_1220_1250_765 (N4-2)**
955_1220_1250_780 (Training)
955_1220_1250_795 (Val)
**955_1220_1250_810 (N4-3)**
955_1220_1250_825 (Training) |

Table 20: Benchmark data details (abnormal configurations).

| Abnormal Behaviour Configurations/ Arbitrary SPs | | | | |
|---|---|---|---|---|
| **Type 1 (start@955-incr-dec/const)** | **Type 2 (start@1220-incr-dec)** | **Type 3 (start@1220-dec-inc)** | **Type 4 (start@1250-dec-inc)** | **Type 5 (start@1250-dec-inc)** |
| 955_1220_1200_750.csv (Training)
955_1220_1210_750.csv (Val)
955_1220_1220_750.csv
955_1250_1220_750.csv (Training)
955_1250_1220_765.csv (Val)
955_1250_1250_750.csv
955_1260_1250_750.csv (Training)
955_1270_1250_750.csv | 1220_1250_955_750.csv (Training)
1220_1250_955_795.csv | 1220_955_1250_750.csv (Training)
1220_955_1250_780.csv | 1250_955_1220_750.csv (Training)
1250_955_1220_825.csv | 1250_1220_955_750.csv (Training)
1250_1220_955_810.csv |

Table 20 details the remaining 16 configurations representing abnormal conditions (arbitrary set points). These are split for training and validation to make the model robust during training (similar to adversarial learning). We set aside 7 configurations apart from training/validation. A well-trained physics-aware model should perform poorly on these, rendering them unnecessary for testing.

For training a DL model, we aggregate the configuration datasets belonging to training splits as shown in Table 19. Prior to collecting, each of the datasets are reformatted to obtain time-shifted input-output pairs as discussed in the data generation methodology. After that rows of these training datasets are shuffled and stacked together to train the model. Each configuration is stored by a .csv file containing 1500 time steps sampled with a 15s delay, to account for conduction analysis. Thus, each configuration accounts for 6.25h worth data. Considering all 50 datasets, our generated data sets consists of 312.5h (or roughly, 13 days) of furnace data. We observed diminishing returns on model

performance with further data size increases, justifying our decision to focus on this efficient data volume.

During time-shifted input-output pairs formation from a configuration dataset, we drop the first and last rows resulting in 1498 rows, to account for the shift operations. Thus, by consolidating the 20 training datasets, we get a total of 29960 train rows. These can be packed within a standard DataLoader in a framework like PyTorch, and train an off-the-shelf DL model. We can similarly obtain 17976 val rows, and also 26964 test rows (from across normal and abnormal configurations, if desired). We have reported results on the 11 datasets individually, where a model trained is used for auto-regressive, sequential prediction of subsequent time steps.

The discussed data sets, along with necessary data pre-processing, model training/evaluation scripts are provided in the following github repository <https://github.com/>, which shall be updated periodically to reflect the latest changes as available (while adhering to FAIR guidelines (Wilkinson et al., 2016)). As a highlight, we provide the *configuration datasets* as separate .csv files. We also provide the consolidated stacked data as a .npz file. Furthermore, we also provide the TEA data as individual files, which are used during model training.

## A.9  POTENTIAL REAL-LIFE APPLICATIONS OF THE WORK AND ITS IMPACT

We now discuss how our method for furnace temperature profiling can be applied in various industries and contribute to energy efficiency and reduced emissions.

**Steel and Metal Manufacturing**: Our model can be directly applied to improve the efficiency of reheating furnaces used in steel and metal manufacturing processes. By providing accurate real-time temperature predictions, operators can optimize fuel consumption and reduce energy waste, leading to significant cost savings and lower carbon footprint. The ability to precisely control temperature profiles can also enhance product quality and consistency.

**Glass and Ceramic Production**: In the glass and ceramic industries, furnaces are crucial for melting, annealing, and tempering processes. Our model can be adapted to these furnace types, enabling tighter temperature control, reduced energy usage, and minimized defects. This can translate to higher productivity, lower operational costs, and a greener manufacturing process.

**Cement and Lime Production**: High-temperature furnaces are essential in cement and lime manufacturing for calcination and clinker production. Our physics-aware deep learning approach can be leveraged to optimize these processes, reducing fuel consumption and emissions while maintaining product quality. This can contribute to the sustainability efforts of cement and lime producers.

**Petrochemical Refining**: Furnaces are widely used in petrochemical refineries for various processes such as crude oil distillation, catalytic cracking, and reforming. By implementing our model, refineries can enhance energy efficiency, minimize fuel wastage, and lower greenhouse gas emissions. This can help refineries meet stringent environmental regulations while maintaining profitability.

## A.10  LIMITATIONS AND FUTURE WORK

**Incorporation of Geometry-Specific Regularization**: Future research should investigate the integration of geometry-specific regularization terms into our model. This could involve developing customized regularization strategies that account for the unique thermal characteristics of various furnace designs. By tailoring the model to specific configurations, we can potentially enhance its predictive accuracy and applicability across different industrial scenarios. This is beyond the scope of our work, which could be treated as a starting point in this direction.

**Exploration of Foundational Models**: Our approach could serve as a foundation for developing models that can be adapted for other related use cases. We envision leveraging techniques such as few-shot learning, continual learning, or transfer learning to enable our model to learn from limited data in new contexts. This would allow for rapid adaptation to different operational conditions and requirements, making our model more versatile and applicable across various industries.

**Engineering aspects of Integration with Real-Time Monitoring Systems**: Extensive study of challenges involved during engineering integration in a monitoring system could itself be another future direction of study, especially for a varied set of industries and furnace configurations.

