# OpenReview forum: "Hottel Zone Physics-Constrained Networks for Furnaces"
_ICLR.cc/2025/Conference — ICLR 2025 Conference Withdrawn Submission_

### Official Review · Reviewer_Whrm · 2024-10-29

**Soundness:** 3
**Presentation:** 1
**Contribution:** 3
**Rating:** 5
**Confidence:** 3

**Summary:**

The manuscript presents a physics-aware deep learning model that integrates principles from the Hottel zone method for real-time prediction of furnace temperatures in the foundation industry. The authors propose a reformulation of the Hottel zone method's equations and introduce an energy-balance based constraint as a regularizer within the neural network. Results are demonstrated across various neural network architectures, showcasing the effectiveness of their approach.

**Strengths:**

The problem is well motivated and is significant for the foundation industry.
The use of physics-based constraints using principles of conventional zone method appears to be a good idea and is also shown to enhance the quality of the predictions. Such efforts are vital and essential for sectors such as foundation industry, where much work is needed on adoption of machine learning and deep learning technologies. This could represent a valuable applied contribution to the process industry.

**Weaknesses:**

Clarity:
Structure of the paper makes reading quite difficult. Important information for evaluation is dispersed throughout the paper at varied locations, including a quite lengthy appendix. A more organized structure would enhance the clarity of the paper. Consider compressing the important information in tables wherever possible, instead of paragraphs.

Originality/Novelty:
1. The claim of being "first-of-its-kind" is questionable in light of a paper presented at NeurIPS 2023 workshop (https://arxiv.org/abs/2308.16089), which has not been cited. What is the improvement achieved over earlier results? In contrast, the results reported here appear to be inferior going by the scale of error metric values. A comparative analysis with the mentioned work would be beneficial to clarify this discrepancy.
2. Investigation of Physics-constraints in variety of neural network architecture does not add value to the paper. The reason of choosing these specific network architectures is also not clear.
3. Hottel zone method seems to take only 5 minutes for simulating temperatures across entire furnace for a 341 minute real process. For 1 time step, the PCNN model takes 0.5 seconds. What is the total time taken by PCNN for 341 minute process prediction? From a practical standpoint, is it even worth developing the PCNN model that accelerates the prediction marginally while losing accuracy?

Finally, while this work is an important piece of applied ML work, it does not qualify as an original contribution to the field of machine learning in reviewer's opinion.

**Questions:**

1. Have authors thought about using advanced neural operator frameworks for solving this problem? E.g. Fourier Neural Operators, Deep Neural Operators.

---

### Official Review · Reviewer_qbGN · 2024-10-30

**Soundness:** 3
**Presentation:** 1
**Contribution:** 2
**Rating:** 5
**Confidence:** 3

**Summary:**

This paper investigates a novel approach to improve the temperature profile prediction of furnaces in foundation industries. Physics-constrained networks are used to obtain the real-time inference capabilities and careful generalization for real-world applications
The effectiveness on various neural network architectures, including Multi-Layer Perceptrons (MLP), Long Short-Term Memory (LSTM),
Extended LSTM (xLSTM) and Kolmogorov-Arnold Networks (KANs) are demonstrated.

**Strengths:**

This works conducts many experiments on different datasets and different methods to evaluate the model.

**Weaknesses:**

The organization of this paper should be improved. The results from MLP, LSTM, DLSTM, KAN and xLSTM and their physics-based improvements (PBMLP, PBLSTM, PBDLSTM, PBKAN and PBxLSTM) are compared. But this paper is to improve the neural networks using Hettel Zone constraint, I think the results from pure Hettel Zone method should also be included. If it is a matter of pages limit, I suggest to move some text of the background of the proposed method (in Section 3) to the appendix. Other than that, the author should consider does it really neccessary to compare the results from so many neural networks with simple structures. Adding physics terms in loss function is a normal method, like in continuum fluid mechanics, so this is an application of PINN in furnace temperature prediction. The author should also check the grammar, such as the last sentence of the Abstract ("We also discussion the data generation involved").

**Questions:**

As mentioned in the "Weaknesses" part, I suggest the authors reganize the paper and focus on several issues due to the limit of pages.

---

### Official Review · Reviewer_96th · 2024-10-31

**Soundness:** 3
**Presentation:** 3
**Contribution:** 2
**Rating:** 5
**Confidence:** 3

**Summary:**

This paper proposes a physics-constrained neural network approach for temperature profile prediction in furnaces. The authors introduce a regularization technique based on the Hottel Zone method and demonstrate its effectiveness across various neural network architectures (MLP, LSTM, KAN, xLSTM).

**Strengths:**

**Originality**
   This work presents an innovative application of the Hottel Zone method for regularizing neural networks, which sets it apart from traditional data-driven models. Using a physics-based constraint to enhance temperature prediction accuracy is a novel concept.

**Clarity**
   The methodology, especially the integration of the Hottel Zone method, is well-explained, allowing readers to understand how the physics constraint aids in capturing the furnace's temperature dynamics. Experimental design and metrics are clearly presented.

**Weaknesses:**

**Domain-Specific Limitation**
   The method is specifically tailored to furnace temperature prediction, limiting its general applicability to other domains. This specificity reduces its potential impact within a broader range of applications or datasets outside of high-temperature industrial processes.

**Inference Time Analysis**
   Although training time with regularization is detailed, the paper lacks explicit performance analysis for inference time on industrial setups. Given the potential complexity of incorporating physical constraints, a clear discussion on inference time efficiency for real-time applications would strengthen the paper's practical value.

**Comparative Baseline**
   The paper does not provide a comprehensive comparison with other physics-informed neural networks or hybrid models, which makes it difficult to evaluate the novelty of this approach relative to existing techniques in similar settings.

**Implementation Complexity**
   The integration of Hottel Zone constraints may complicate the model deployment process in practical, real-time environments. The paper would benefit from discussing strategies to mitigate the computational overhead associated with this regularization approach.

**Questions:**

1. How does the proposed regularization technique affect inference time? Is it feasible for real-time applications, particularly in industrial settings? Could you provide specific inference time benchmarks or comparisons to baseline models without the physics-based regularization?

2. Could the physics-constrained regularization approach be adapted for other high-temperature processes or domains where similar physical constraints exist, such as chemical reactors or solar power systems? What modifications, if any, would be required? Could you discuss which aspects of your approach are furnace-specific versus which could be more easily adapted to other domains?

3. The paper distinguishes this approach from typical PINNs. Could the authors elaborate on specific advantages of their approach over traditional PINNs in similar scenarios?

4. Has the proposed model been tested in actual industrial furnace setups? If not, what challenges or adaptations would be expected in deploying it for real-time industrial applications?

---

### Official Review · Reviewer_4N4c · 2024-11-04

**Soundness:** 1
**Presentation:** 1
**Contribution:** 1
**Rating:** 3
**Confidence:** 3

**Summary:**

This paper studies temperature profile prediction of furnaces and tries to integrate a typical approach named Hottel Zone into Deep Learning models.

**Strengths:**

While I may lack specific expertise in furnace behaviors, the paper’s strengths are difficult to identify due to significant weaknesses in structure and clarity.

**Weaknesses:**

The primary weakness of this paper is its lack of a clear, structured approach, which hampers readability and information flow. Also, the writing style is very unclear and there are missing unfinished statements, links (like the GitHub from line 68), the red text, etc.

**Questions:**

I recommend that the authors rewrite and restructure the paper to improve its readability and coherence

**Details Of Ethics Concerns:**

It could be that this paper was completely generated by an LLM.

---

### Note · Authors · 2024-11-16

**Comment:**

The reviewers have provided shallow comments w/o reading the paper well. All of them seem to have no experience in the domain of the paper. Providing a rebuttal does not make sense given the pool of inexperienced comments.

**Withdrawal Confirmation:**

I have read and agree with the venue's withdrawal policy on behalf of myself and my co-authors.